# Vibration-Based Recognition of Wheel–Terrain Interaction for Terramechanics Model Selection and Terrain Parameter Identification for Lugged-Wheel Planetary Rovers

**DOI:** 10.3390/s23249752

**Published:** 2023-12-11

**Authors:** Fengtian Lv, Nan Li, Haibo Gao, Liang Ding, Zongquan Deng, Haitao Yu, Zhen Liu

**Affiliations:** 1State Key Laboratory of Robotics and System, Harbin Institute of Technology, Harbin 150001, China; lvfengtian@sia.cn (F.L.); gaohaibo@hit.edu.cn (H.G.); liangding@hit.edu.cn (L.D.); dengzq@hit.edu.cn (Z.D.); yht@hit.edu.cn (H.Y.); zhenliu.hit@gmail.com (Z.L.); 2State Key Laboratory of Robotics, Shenyang Institute of Automation, Chinese Academy of Sciences, Shenyang 110016, China; 3Institutes for Robotics and Intelligent Manufacturing, Chinese Academy of Sciences, Shenyang 110169, China; 4Key Laboratory of Marine Robotics, Liaoning Province, Shenyang 110169, China

**Keywords:** terrain parameter identification, wheel–terrain interaction classes, vibration features, terrain classification, planetary rover

## Abstract

Identifying terrain parameters is important for high-fidelity simulation and high-performance control of planetary rovers. The wheel–terrain interaction classes (WTICs) are usually different for rovers traversing various types of terrain. Every terramechanics model corresponds to its wheel–terrain interaction class (WTIC). Therefore, for terrain parameter identification of the terramechanics model when rovers traverse various terrains, terramechanics model switching corresponding to the WTIC needs to be solved. This paper proposes a speed-independent vibration-based method for WTIC recognition to switch the terramechanics model and then identify its terrain parameters. In order to switch terramechanics models, wheel–terrain interactions are divided into three classes. Three vibration models of wheels under three WTICs have been built and analyzed. Vibration features in the models are extracted and non-dimensionalized to be independent of wheel speed. A vibration-feature-based recognition method of the WTIC is proposed. Then, the terrain parameters of the terramechanics model corresponding to the recognized WTIC are identified. Experiment results obtained using a Planetary Rover Prototype show that the identification method of terrain parameters is effective for rovers traversing various terrains. The relative errors of estimated wheel–terrain interaction force with identified terrain parameters are less than 16%, 12%, and 9% for rovers traversing hard, gravel, and sandy terrain, respectively.

## 1. Introduction

Planetary rovers need to traverse rough terrains during planetary exploration missions. Wheel–terrain interaction has substantial influence on the mobility of rovers [1,2]. The typical surfaces of Mars consist of hard, sandy, and gravel surfaces [3]. Rovers can move quickly on hard, flat, and non-slippery surfaces. However, they are typically required to traverse over sandy areas, where they suffer from wheel sinkage, or over gravel surfaces where violent vibrations may happen. Hazards originating from the terrain surface itself are called non-geometric hazards. Knowledge of the terrain class and other properties can help improve the mobility of the rover by adjusting its path or control strategy in advance. Recognition of terrain class, terramechanics modeling, and terrain parameter identification are essential to optimize the control of high-mobility rovers. Research progress and existing problems related to terrain classification and parameter identification are discussed in this section.

Terrain classification can be divided into exteroceptive and proprioceptive terrain classification. Exteroceptive terrain classification mainly applies vision or range data [4,5,6,7,8]. For example, Otsu et al. [4] used semi-supervised algorithms instead of supervised algorithms for visual classification systems. Support vector machine (SVM) is used to classify data from different sensors. For the excellent performance of deep learning in object classification and recognition, some studies have utilized deep learning algorithms for terrain classification of planetary rovers, achieving high classification accuracy [6,7]. However, the vision- or laser-based classification methods are usually related to the topmost terrain surface, which may not be the load-bearing surface. An example of this is the Mars terrain, which is covered by a thin layer of drift sand.

Proprioceptive terrain classification has focused on vibration-based terrain classification methods [9,10,11,12,13,14]. Brooks and Iagnemma [9] extracted the power spectral density (PSD) of vibration acceleration to classify the terrain into gravel or sand. In a later work, vibration, vision, and wheel–terrain interaction force were combined to construct a self-supervised terrain classification framework for planetary rovers [10]. However, the terramechanics model used in their paper is not suitable for calculating the interaction force of the wheel and hard/gravel terrain. The terrain parameters used in the terramechanics model are also unidentified. Vicente et al. [11] extracted eleven statistical features of vibration accompanied with the PSD of vibration acceleration to form a new feature vector. An extreme learning machine was used to classify terrains based on the feature vector. Some studies applied deep learning algorithms to classify planetary terrain based on proprioceptive sensors [12,13]. Those methods can reduce classification accuracy caused by inappropriate feature extraction and selection. These methods neglect that the variation of the wheel speed (over a given terrain) changes the amplitude and frequency of the measured wheel vibration signal. If rovers move on the same class of terrain at different speeds, the class of terrain may be incorrectly classified. One solution is to increase the number of trained speeds to obtain vibration features at various speeds [14]. However, there is a drawback that numerous samples at the rover’s multiple moving speeds need to be obtained. Another solution is to extract speed-independent vibration features of the wheel that are proposed in this paper.

Terrain parameters are useful for planetary rovers to maximize the wheel traction or minimize the power consumption [15]. Several methods have been reported to estimate terrain parameters [16,17,18,19]. Iagnemma et al. [16] presented a linearized terramechanics model for estimating the internal cohesion and friction angle of soil. However, many other important terrain parameters such as the cohesive modulus, friction modulus, and sinkage exponent in terramechanics models have not been identified. Hutangkabodee et al. [17] identified the parameters, including the internal friction angle, shearing deformation modulus, and lumped pressure sinkage coefficient, using the Newton–Raphson method based on a simplified terramechanics model. Ding et al. presented an improved terramechanics model with consideration of the slip–sinkage and lug effects. Soil parameters were divided into three independent sets by decoupling a wheel–soil terramechanics model. This model can be used to identify Lunar terrain parameters with the Yutu-2 rover [18]. However, these methods cannot be applicable to rovers traversing firm soil because the assumed terramechanics models do not cover interaction between the rover wheels and hard terrain (HT). The soil of HT does not experience a shear deformation, and there is no shear stress in the wheel–terrain interaction. The soil cohesion coefficient (*c*) or shearing deformation modulus (*K*) is not part of the shear characteristic parameters in the shear terramechanics model of the interaction between a wheel and HT. To address this problem, Yang et al. [19] set parameters *K* and *c* equal to zero to make a terramechanics model of the interaction between a wheel and soft terrain (ST) suitable for the interaction between the wheel and HT. It is necessary to carry out terrain classification to determine parameters such as *K* and *c* in a terramechanics model when calculating interaction force or torque.

Conventional terrain classification classifies terrain classes according to their components to help rovers and mobile robots adjust their control strategy and path planning through which a qualitative judgment of the terrain properties is acceptable. However, an autonomous planetary rover should “know” the terrain properties to calculate the wheel–terrain interaction force to maximize the wheel traction or minimize the power consumption and improve the traversability of the rover during its operational phase. Despite efforts in this regard, the terrain parameter identification of various terrain classes and terramechanics model switching for the interaction between the wheel and soft or hard terrain are still issues to be researched for planetary rovers.

As mentioned above, the following problems are still associated with research works on terrain classification and parameter identification for planetary rovers: (1) a suitable terramechanics model needs to be switched to for rovers moving on various types of terrain. The recognition of wheel–terrain interaction class (WTIC) has been rarely studied. (2) Parameter identification of terramechanics models in various types of terrain (not just sand) needs to be studied. And (3) reported vibration-based speed-independent terrain classification methods require numerous training samples at multiple speeds, which is not feasible for planetary rovers. To solve these problems, the following need to be investigated: (1) how to extract speed-independent vibration features and recognize WTIC to determine the corresponding terramechanics model, especially in the regions where the terramechanics model needs to be switched, and (2) how to identify parameters in the terramechanics model for a planetary rover traversing various types of terrain.

In this paper, the speed-independent vibration features are extracted to recognize WTIC first. Terrain parameters in the terramechanics model that is switched to corresponding to the recognized WTIC are then identified. This paper comprehensively utilizes various sensors installed on the planetary rovers, including accelerometers, cameras, and six-dimensional force/torque sensors, to achieve the recognition of WTIC and identification of terrain parameters. The accelerometer is used to collect wheel vibration acceleration data. Speed-independent vibration features are extracted from the acceleration data to recognize WTIC and select the corresponding terramechanics model. The camera can obtain wheel–terrain interaction images to realize the estimation of wheel sinkage and slip ratio using machine vision technology. The wheel–terrain interaction force information of the rover wheel is acquired using the six-dimensional force/torque sensor. Then, the wheel–terrain interaction force, wheel sinkage and slip ratio are input into the terramechanics model to identify terrain parameters.

The main contributions of this paper are as follows: (1) corresponding to the three wheel–terrain interaction classes (WTICs), this paper proposes three vibration models of lugged wheels and simplifies them to be approximated as periodic functions, showing the difference in wheel vibration frequency and amplitude for different WTICs. Through the dimensionless processing of the amplitude, frequency, mean, and mean square value of the vibration acceleration, four speed-independent vibration features are extracted to reflect the influence of the terrain on the vibration of the wheel. (2) A vibration-based WTIC recognition method without requirement for training data is proposed. Results of single-wheel test experiments show that the simple and effective method has recognized 292 samples of 300 samples correctly, so the recognition accuracy is about 97%. (3) Corresponding to the recognized WTIC, the terramechanics model is switched by determining whether the parameters of the shearing deformation modulus (*K*) and soil cohesion coefficient © in the model are 0. Terrain parameters in the terramechanics model are identified using the least squares method for rovers traversing various types of terrain. Experiment results obtained using a Planetary Rover Prototype moving on three terrain classes show that the proposed vibration-based WTIC recognition method is effective and has recognized 411 samples of 432 samples correctly, and the terrain parameter identification for various terrains is also effective.

The remainder of this paper is organized as follows. Section 2 discusses the vibration models of the wheel for different WTICs. Based on these models, speed-independent features extracted from the vibration acceleration are described in Section 3. Section 4 presents a method for WTIC recognition by analyzing the range of vibration features for different WTICs. In Section 5, a method is proposed for identifying parameters of the terramechanics model in various types of terrain. The method switches the corresponding terramechanics model according to the recognized WTIC and then uses the model to identify terrain parameters. Section 6 discusses the experiment results of the WTIC recognition and parameter identification for a Planetary Rover Prototype. Finally, some concluding remarks are provided in Section 7.

## 2. Vertical Vibration for Lugged Wheels of Planetary Rovers

### 2.1. Vertical Dynamics Model of Wheel Motion

A lugged wheel can be modeled as a mass–spring–damper system shown in Figure 1 [20], which is composed of the mass *m_w_* that the wheels of the rover need to load, a spring with a stiffness coefficient *k_w_*, and a damper with a damping coefficient *c_w_*. Here, *y*(*t*) is the vertical output displacement, and *q*(*t*) is the base excitation, which is a time-varying vertical displacement of the lowest point of the wheel rim. In addition, *Y*(*jω*) and *Q*(*jω*) are the Fourier transforms of *y*(*t*) and *q*(*t*), respectively, where *Y*(*jω*) = *H*(*jω*)*Q*(*jω*). The frequency response function *H*(*jω*) is as follows:(1)$$H\left(j\omega \right)=\left(c_{w}j\omega +k_{w}\right)/\left(-m_{w}\omega^{2}+c_{w}j\omega +k_{w}\right),$$

The modulus of *H*(*jω*) can be expressed as:(2)$$\left|H\left(j\omega \right)\right|=\sqrt{\left(1+{\left(2\xi \lambda \right)}^{2}\right)/\left({\left(1-\lambda^{2}\right)}^{2}+{\left(2\xi \lambda \right)}^{2}\right)},$$
where $\xi =c_{w}/2\sqrt{k_{w}m_{w}}$ is the damping ratio, $\lambda =f_{v}/\omega_{n}$ is the frequency ratio, $f_{v}$ is the frequency of base excitation *q*(*t*), and $\omega_{n}=\sqrt{k_{w}/m_{w}}$ is the natural frequency of the undamped free mass–spring–damper system. According to [20], $\left|H\left(j\omega \right)\right|$ is approximately equal to 1 when λ is less than 0.75.

For a lugged wheel on the rover, the spring stiffness *k_w_* is very big. The mass *m_w_* is assumed as 29 kg, the same as the mass of Spirit rover equally distributed to each wheel. The rover always travels at a low speed, for example, the speed of the Spirit rover is less than 50 mm/s. As a result, the natural frequency *ω_n_* is much bigger than the frequency $f_{v}$ of base excitation that is an approximate periodic function for a lugged wheel, as shown later in this paper. Therefore, λ is less than 0.75 and $\left|H\left(j\omega \right)\right|$ is approximately equal to 1. Hence,
(3)$$Y\left(j\omega \right)\approx Q\left(j\omega \right),$$

According to the properties of a Fourier transform, the following approximation can be obtained.
(4)$$y\left(t\right)\approx q\left(t\right).$$

Therefore, the vertical output displacement and base excitation of the rigid lugged wheel can be regarded as equal in value, i.e., $y\left(t\right)=q\left(t\right)$. Thus, in the following section, the vibration base excitations of the wheel moving on various terrain classes are analyzed to obtain the wheel vibration output displacements.

### 2.2. Vibration Models for Wheels Moving on Different Terrains in Wheels’ Center Plane

Wheels with lugs parallel to the wheel axis are usually used on planetary rovers, such as Spirit rover’s wheels. This type of wheel is studied in this paper. The vibrations of this type of wheel result from lug motion and terrain roughness/hardness. Terrains are mainly divided into soft and hard terrain. Both of them can be divided into low-unevenness and high-unevenness terrain. When the planetary rover travels on different classes of terrain, the terramechanics model is different and needs to be switched by setting the values of terrain parameters *K* and *c* in the model. In order to determine the values of *K* and *c*, wheel–terrain interactions are divided into three classes as shown in Figure 2: lugs entering into soil (LEIS), all lugs contacting terrain surface (ALCT), some lugs contacting terrain surface (SLCT). When the wheel moves on low-unevenness soft terrain and high-unevenness soft terrain, every lug can enter into the soil. When the wheel moves on low-unevenness hard terrain, every lug can contact the terrain surface. When the wheel moves on high-unevenness hard terrain, the same lugs can contact the terrain surface while other lugs cannot contact with the terrain for a wheel rotating 360°. The terrain is classified into three groups: the first class of terrain (T1), which corresponds to WTIC 1: ALCT, prevents each lug from entering the soil during wheel movement, although each lug can be in contact with the terrain. The second class of terrain (T2), which corresponds to WTIC 2: SLCT, makes it impossible for the lugs to enter the soil during the wheel movement, and some lugs cannot be in contact with the terrain. The third class of terrain (T3), which corresponds to WTIC 3: LEIS, refers to the terrain when the wheels transverse it, and where the lugs can enter the soil. The correspondence between terrain classes and the WTICs is shown in Table 1.

The origin of coordinate system Σ*o*-*xy* is located at the wheel center when the wheel is stationary. The *x*-axis is parallel to the horizontal direction, and the *y*-axis is opposite the gravity direction. Σ*o*-*xy* does not move with the wheel. The angle velocity of the wheel is *ω_w_*, *l* is the lug height, *n_l_* is the number of lugs, and *α_nl_* (*α_nl_* = 2π/*n_l_*) is the angle between two adjacent lugs.

According to the number of lugs on the wheel, the wheel can be divided into two classes: olig-lugged and multi-lugged wheels. An olig-lugged wheel is defined as the lugged wheel whose rim can contact the terrain surface when traversing hard terrain (HT). A multi-lugged wheel, which is often used on rovers, is defined as the wheel equipped with a large number of lugs that keep its rim from making contact with the terrain surface when traversing HT.

#### 2.2.1. Vibration Model of Wheels Moving on Flat Hard Terrain

Flat hard terrain is regarded as the reference terrain. The vertical vibration of a wheel traversing the reference terrain is defined as the vertical reference vibration.

Figure 3 shows the movement phases for a wheel rotating around a supporting lug on a reference terrain. Here, *θ* (−*α_nl_*/2 < *θ* ≤ *α_nl_*/2) is the angle from the direction of the support lug to the gravity direction at the initial time when the wheel starts to move in the anticlockwise direction, which is taken as the positive direction of the angle. At any time, there is no more than one lug in contact with the terrain when an olig-lugged wheel traverses the reference terrain. Assume *θ* is equal to *α_nl_*/2 at the initial time *t*_1_ when the wheel starts to move. The movement of the olig-lugged wheel rotating around the support lug can be divided into three phases. (1) Wheel center rising phase: this phase starts at moment *t*_1_, when the support lug starts to be in contact with the terrain, until moment *t*_2_ when the support lug is perpendicular to the terrain. The wheel center moves from the lowest point to the highest point along the *y*-axis. (2) Wheel center dropping phase: this phase starts at moment *t*_2_ until moment *t*_3_ when the wheel rim starts to be in contact with the terrain. During this phase, the wheel center moves from the highest point to the lowest point along the *y*-axis. (3) Wheel center height unchanging phase: this phase is from moment *t*_3_ to moment *t*_4_ when the next adjacent lug starts to be in contact with the terrain. The height of the wheel center maintains the lowest invariance.

The wheel repeats the above movement phases with the lugs contacting the terrain in turn. The distance *h_rt_*(*t*) between the lowest point of the wheel rim and the reference terrain surface can be expressed as follows:(5)$$h_{rt}(t)=\max\left(0,(r+l)\cos\left(\varphi_{rt}(t)\right)-r\right).$$
where *r* is the wheel radius, *l* is the lug height, and $\varphi_{rt}(t)$ is the angle between the support lug and gravity direction at the moment *t*. It can be expressed as
(6)$$\varphi_{rt}(t)=\theta -\omega_{w}t+\alpha_{nl}\times \mathrm{INT}\left(\left(\omega_{w}t+\alpha_{nl}/2-\theta \right)/\alpha_{nl}\right).$$
(7)$$\begin{array}{c} \mathrm{The}\;\mathrm{wheel}\;\mathrm{base}\;\mathrm{excitation}\;\mathrm{is}\;q_{\mathrm{rt}}(t)=h_{\mathrm{rt}}(t)-\min(h_{\mathrm{rt}}(t))=(t).\;\mathrm{Based}\;\mathrm{on}\;\\ \mathrm{Equation}\;(5),\;\mathrm{the}\;\mathrm{vertical}\;\mathrm{output}\;\mathrm{displacement}\;\mathrm{of}\;\mathrm{the}\;\mathrm{wheel}\;\mathrm{is}\\ y_{rt}(t)=q_{rt}(t)=\max\left(0,(r+l)\cos\left(\varphi_{rt}(t)\right)-r\right). \end{array}$$

For a multi-lugged wheel, the movement of the wheel rotating around the supporting lug can be divided into wheel center rising phase and dropping phase. The wheel center height unchanging phase, in which the wheel rim can be in contact with the terrain, does not exist. The distance *h_rt_*(*t*) is as follows:(8)$$h_{rt}(t)=\left(r+l\right)\cos\left(\varphi_{rt}(t)\right)-r.$$

The output displacement of the wheel is:(9)$$y_{rt}\left(t\right)=q_{rt}\left(t\right)=\left(r+l\right)\cos\left(\varphi_{rt}(t)\right)-\cos\left(\alpha_{nl}/2\right)\left(r+l\right).$$

When the multi-lugged wheel traverses the reference terrain, there are no more than two lugs in contact with the terrain, and the wheel rim cannot be in contact with the terrain surface. Thus, $(r+l)\cos\left(\varphi_{rt}(t)\right)-r$ is bigger than zero. The *h_rt_*(*t*) calculated using Equation (8) is equal to *h_rt_*(*t*) calculated using Equation (5). Equation (8) can be regarded as a special case of Equation (5). Whether the lugs on the wheel are olig or multi, the output displacement for a lugged wheel traversing the reference terrain can be expressed uniformly as follows:(10)$$y_{rt}(t)=q_{rt}(t)=h_{rt}(t)-\min(h_{rt}(t))=\max\left(0,(r+l)\cos\left(\varphi_{rt}(t)\right)-r\right)-\max\left(0,\cos\left(\alpha_{nl}/2\right)\left(r+l\right)-r\right).$$

#### 2.2.2. Vibration Model of Wheels with All Lugs Contacting Terrain Surface

In an actual situation, the surface of HT is rugged, as shown in Figure 4. The class of wheel–HT interaction is ALCT. The wheel movement on the terrain is similar to the wheel movement on a reference terrain. The characteristics of the wheel movement are irrelative to the state of the wheel at the initial time. Thus, this paper assumes that the lug starts to contact the terrain at the initial moment *t*_1_. The terrain coordinate system *O*-*XY* is located at the terrain surface. The *X*-axis is parallel to the horizontal direction, and the *Y*-axis coincides with the *y*-axis. The roughness function of the terrain is expressed as *g*(*X*). The distance *h*(*t*) between the lowest point of the wheel rim and the terrain surface is
(11)$$h(t)=\max\left(g\left(S_{x}(t)\right),\left(r+l\right)\cos\left(\varphi \left(t\right)\right)+g\left(X_{n}\right)\right).$$
(12)$$S_{x}(t)=X_{n}-\left(r+l\right)\sin\left(\varphi \left(t\right)\right).$$
(13)$$\varphi (t)=\theta -\omega_{w}t+\left(n\left(t\right)-1\right)\alpha_{nl}.$$
where *S_x_*(*t*) is the wheel displacement along the *x*-axis, *n* is the number of times the lugs contact the terrain from *t*_1_ to the moment *t*, *X_n_* is the position of the supporting lug on the *x*-axis.
(14)$$\left\{\begin{array}{l} X_{1}=\left(r+l\right)\cos\theta \\ X_{n+1}=X_{n}+a_{1}\left(r+l\right)+\omega_{w}r(t_{e}{}_{n}-t_{s}{}_{n}) \end{array}\right..$$
(15)$$a_{1}=\left(\sin\left(\varphi \left(t_{s}{}_{n}\right)\right)+\sin\left(\varphi \left(t_{e}{}_{n}\right)+\alpha_{nl}\right)\right).$$
where $t_{s}{}_{n}$ is the moment when the supporting lug starts to depart from the terrain, and $t_{e}{}_{n}$ is the moment when the next lug starts to contact the terrain.

During the movement of the wheel, the position of the lug adjacent to the supporting lug on the *X*-axis is
(16)$$X_{nl}(t)=\left\{\begin{array}{l} X_{n}+\left(r+l\right)\times \eta_{1}\left(t\right)\;\;\;\;\;\;t_{e}{}_{\left(n-1\right)}\leq t\leq t_{s}{}_{n}\\ X_{n}+a_{1}\left(r+l\right)+\omega_{w}r(t-t_{s}{}_{n})\;t_{s}{}_{n}\leq t\leq t_{e}{}_{n} \end{array}\right..$$
(17)$$\eta_{1}\left(t\right)=\sin\left(\varphi \left(t\right)\right)\left(\cos\alpha_{nl}-1\right)+\cos\left(\varphi \left(t\right)\right)\sin\alpha_{nl}.$$

The distance between the lug adjacent to the supporting lug and the terrain surface is
(18)$$\tilde{h}\left(t\right)=h\left(t\right)-h^{\prime }\left(t\right).$$
(19)$$h^{\prime }\left(t\right)=\left(r+l\right)\cos\left(\varphi \left(t\right)+\alpha_{nl}\right)+g\left(X_{nl}\left(t\right)\right).$$
where $h^{\prime }\left(t\right)$ is the sum of the distance between the lug adjacent to the supporting lug and wheel center and terrain roughness of position $X_{nl}\left(t\right)$ on the *x*-axis.

Then, *n* can be expressed as follows:(20)$$n(t^{+})=\left\{\begin{array}{l} \mathrm{INT}\left(\alpha \left(t\right)/\alpha_{nl}\right)\times \mathrm{INT}\left(h^{\prime }/h\right)\;\alpha \left(t\right)\geq (n+1)\alpha_{nl}\\ \mathrm{INT}\left(\alpha \left(t\right)/\alpha_{nl}\right)+\mathrm{INT}\left(h^{\prime }/h\right)\;\alpha \left(t\right)<(n+1)\alpha_{nl} \end{array}\right..$$
(21)$$\alpha \left(t\right)=\omega_{w}t+\alpha_{nl}/2-\theta .$$
where *t*^+^ is the moment next to *t*, and *n*(0) is equal to 1. The vertical displacement of the wheel is
(22)$$y\left(t\right)=q\left(t\right)=h\left(t\right)-\min\left(h\left(t\right)\right).$$

For a multi-lugged wheel,
(23)$$X_{nl}(t)=X_{n}+\left(r+l\right)\times \eta_{1}\left(t\right).$$
(24)$$h(t)=\left(r+l\right)\cos\left(\varphi \left(t\right)\right)+g\left(X_{n}\right).$$
(25)$$h_{rt}(t)=\left(r+l\right)\cos\left(\varphi_{rt}(t)\right)-r.$$

When a multi-lugged wheel traverses HT, since the wheel rim does not contact the terrain, $t_{s}{}_{n}$ is equal to $t_{e}{}_{n}$ and $\left(r+l\right)\cos\left(\varphi \left(t\right)\right)+g\left(X_{n}\right)$ is bigger than $g\left(S_{x}(t)\right)$. Thus, the *h*(*t*) calculated using Equation (24) is equal to *h*(*t*) calculated using Equation (11). Equation (24) can be regarded as a case of Equation (11). Whether the lugs on the wheel are olig or multi, the output displacement for a lugged wheel traversing HT can be expressed uniformly using Equation (22) with *h*(*t*) in Equation (11).

#### 2.2.3. Vibration Model of Wheels with Some Lugs Contacting Terrain Surface

In WTIC 2, SLCT, for example, when the wheel moves on gravel terrain (GT), as shown in Figure 5, the lugs will contact gravel during the movement of the wheel. This leads to some of the lugs hanging in the air without contact with the terrain surface or gravel. The angle between two adjacent supporting lugs is no greater than 90°. The distance between the top point of the *i*-th lug (lu*_i_*), which will be in contact with the terrain, and the terrain surface is determined as follows:
(26)$${\tilde{h}}_{n+i}\left(t\right)=h\left(t\right)-{h^{\prime }}_{n+i}\left(t\right)+r.$$
(27)$${h^{\prime }}_{n+i}\left(t\right)=\left(r+l\right)\cos\left(\varphi \left(t\right)+i\alpha_{nl}\right)+g\left(X_{(n+i)l}\left(t\right)\right).$$
where $X_{(n+i)l}\left(t\right)$ is the position of the *i*-th lug (lu*_i_*), which will be in contact with the terrain on the *X*-axis.

Moreover, the wheel rim may contact the gravel and no lug will be in contact with the terrain some of the time. The minimum distance between the wheel rim and terrain surface is as follows:(28)$${\tilde{h}}_{w}\left(t\right)=h\left(t\right)-{h^{\prime }}_{w}\left(t\right)+r.$$
(29)$${h^{\prime }}_{w}\left(t\right)=\underset{-r\leq x_{w}\leq r}{\max}\left(\sqrt{r^{2}-x_{w}{}^{2}}+g\left(S_{x}\left(t\right)-x_{w}\right)\right).$$
where $S_{x}\left(t\right)$ is the displacement of the wheel center along the *X*-axis, *x_w_* is the position of a point on the wheel rim relative to the wheel center along the *x*-axis. $X_{(n+i)l}\left(t\right)$ and $S_{x}\left(t\right)$ can be calculated using the following equations.
(30)$$X_{(n+i)l}\left(t\right)=\left\{\begin{array}{l} X_{w}+x_{li}\left(t\right)\;\;\;\;{h^{\prime }}_{\max}={h^{\prime }}_{w}\left(t\right)=h(t)\\ X_{n}+\eta_{i}\left(t\right)\left(r+l\right)\;{h^{\prime }}_{\max}\neq {h^{\prime }}_{w}\left(t\right)\;\mathrm{or}\;{h^{\prime }}_{\max}={h^{\prime }}_{w}\left(t\right)\neq h(t) \end{array}\right..$$
(31)$$S_{x}(t)=\left\{\begin{array}{l} -r\sin\left(\varphi \left(t\right)-\alpha_{w}\left(t\right)\mathrm{mod}\alpha_{nl}\right)+X_{w}\;{h^{\prime }}_{\max}={h^{\prime }}_{w}\left(t\right)=h(t)\\ X_{n}-\left(r+l\right)\sin\left(\varphi \left(t\right)\right)\;\;\;\;\;\;{h^{\prime }}_{\max}\neq {h^{\prime }}_{w}\;\left(t\right)\mathrm{or}\;{h^{\prime }}_{\max}={h^{\prime }}_{w}\left(t\right)\neq h(t) \end{array}\right..$$
(32)$${h^{\prime }}_{\max}=\max\left(h\left(t\right),{h^{\prime }}_{n+1}\left(t\right),{h^{\prime }}_{n+2}\left(t\right),\ldots ,{h^{\prime }}_{n+m/4}\left(t\right),{h^{\prime }}_{w}\left(t\right)\right).$$
(33)$$\eta_{i}\left(t\right)=\sin\left(\varphi (t)\right)\left(\cos i\alpha_{nl}-1\right)+\cos\left(\varphi (t)\right)\sin i\alpha_{nl}.$$
(34)$$x_{li}\left(t\right)=\left(r+l\right)\sin\left(\varphi \left(t\right)+i\alpha_{nl}\right)-r\sin\left(\varphi \left(t\right)+\alpha_{w}\left(t\right)\mathrm{mod}\alpha_{nl}\right).$$
(35)$$\left\{\begin{array}{l} X_{0}=\left(r+l\right)\cos\theta \\ X_{n+1}=X_{nl}\left(t_{e}{}_{n}\right) \end{array}\right..$$
(36)$$\alpha_{w}\left(t\right)=\arcsin\left(x_{w\max0}/r\right)-\arcsin\left(\left(X_{n}-S_{x}\left(t\right)\right)/\left(r+l\right)\right).$$
(37)$$X_{w}=S_{x}\left(t\right)+x_{w\max0}.$$
(38)$$x_{w\max0}=\underset{x_{w}}{\arg\max}\left({h^{\prime }}_{w}\left(t\right)\right).$$
where *x_w_*_max0_ is the value of *x_w_* that maximizes ${h^{\prime }}_{w}\left(t\right)$ at moment *t*. The number of wheel lugs in contact with the terrain can be expressed as
(39)$$\begin{array}{l} \left\{\begin{array}{l} \mathrm{iff}\;\mathrm{INT}\left(\max\left({h^{\prime }}_{n+1}\left(t\right),{h^{\prime }}_{n+2}\left(t\right),\ldots ,{h^{\prime }}_{n+m/4}\left(t\right),{h^{\prime }}_{w}\left(t\right)\right)/h\left(t\right)\right)=1\vee \\ \max\left({h^{\prime }}_{n+1}\left(t\right),{h^{\prime }}_{n+2}\left(t\right),\ldots ,{h^{\prime }}_{n+m/4}\left(t\right),{h^{\prime }}_{w}\left(t\right)\right)={h^{\prime }}_{n+i}\left(t\right) \end{array}\right.\Rightarrow n\left(t^{+}\right)=n\left(t\right)+i\\ \left\{\begin{array}{l} \mathrm{iff}\;\mathrm{INT}\left(\max\left({h^{\prime }}_{n+1}\left(t\right),{h^{\prime }}_{n+2}\left(t\right),\ldots ,{h^{\prime }}_{n+m/4}\left(t\right),{h^{\prime }}_{w}\left(t\right)\right)/h\left(t\right)\right)=1\vee \\ \max\left({h^{\prime }}_{n+1}\left(t\right),{h^{\prime }}_{n+2}\left(t\right),\ldots ,{h^{\prime }}_{n+m/4}\left(t\right),{h^{\prime }}_{w}\left(t\right)\right)={h^{\prime }}_{w}\left(t\right) \end{array}\right.\Rightarrow n\left(t^{+}\right)=n\left(t\right)+\mathrm{INT}\left(\alpha_{w}\left(t\right)/\alpha_{\mathrm{nl}}\right) \end{array}.$$
where *t*^+^ is the moment next to *t*, ${h^{\prime }}_{n+i}\left(t\right)$ is the sum of the distance between the *i*-th lug (lu*_i_*) and wheel center and terrain roughness of position $X_{(n+i)l}\left(t\right)$ on the *x*-axis, and *m* is the number of lugs. The distance *h*(*t^+^*) is
(40)$$h\left(t^{+}\right)=\left\{\begin{array}{l} r\cos\left(\varphi \left(t^{+}\right)-\alpha_{w}\left(t^{+}\right)\mathrm{mod}\alpha_{nl}\right)+g\left(X_{w}\right)-r\;{h^{\prime }}_{\max}={h^{\prime }}_{w}\left(t\right)=h(t)\\ \left(r+l\right)\cos\left(\varphi \left(t^{+}\right)\right)+g\left(X_{n}\right)-r\;\;\;\;\;\;{h^{\prime }}_{\max}\neq {h^{\prime }}_{w}\left(t\right)\;\mathrm{or}\;{h^{\prime }}_{\max}={h^{\prime }}_{w}\left(t\right)\neq h(t) \end{array}\right..$$

The distance *h*(0) between the wheel rim and terrain surface at the initial moment *t* = 0 is equal to $\max\left(g\left(S_{x}(0)\right),\left(r+l\right)\cos\left(\varphi \left(0\right)\right)+g\left(X_{0}\right)\right)$. The vertical displacement of the vibration output is
(41)$$y\left(t\right)=q\left(t\right)=h\left(t\right)-\min\left(h\left(t\right)\right).$$

When a multi-lugged wheel moves on GT, some lugs will not be in contact with the surface. Thus, the vertical displacement of the multi-lugged wheel can also be expressed as Equation (41).

#### 2.2.4. Vibration Model of Wheels with Lugs Entering the Soil

In WTIC 3: LEIS, the lugs enter into the soil during wheel movement, for example, when a wheel traverses through sandy terrain (ST), as shown in Figure 6. To facilitate an analysis of the vertical output displacement, assume that a lug starts to enter the soil at the initial moment of the wheel movement. The distance *h*(*t*) between the lowest point of the wheel rim and the terrain surface can be expressed as follows:
(42)$$h\left(t\right)=r-\left(r+l\right)\cos\left(\beta -\omega_{w}t\right)-y_{l}\left(t\right)-\left(r+l\right)\cos\left(\beta \right).$$
where $y_{l}(t)$ is the position of the top point on the lug starting to enter the soil on the *y*-axis, *β* is the angle between the lug starting to enter the soil and gravity direction. The positions of the top points on two adjacent lugs on the *y*-axis are expressed as $y_{l1}(t)$ and $y_{l2}(t)$, respectively. They have the relationship $y_{l2}(t)=y_{l1}(t-\alpha_{nl}/\omega_{w})$ when the wheel–terrain interaction condition, including the terrain properties, slip ratio of the wheel, angular velocity of the wheel, load of the wheel, and so on, remains unchanged. Thus, the position of the top points on the frontmost lug in soil on the *y*-axis at moment *t* is $y_{fl}(t)=y_{l}\left(t-\mathrm{INT}\left(\left(\omega_{w}t\right)/\alpha_{nl}\right)\times \alpha_{nl}/\omega_{w}\right)$. It can be used to calculate the distance *h*(*t*) at moment *t*. Therefore,
(43)$$h\left(t\right)=r-\left(r+l\right)\cos\left(\beta -\omega_{w}t+\mathrm{INT}\left(\left(\omega_{w}t\right)/\alpha_{nl}\right)\times \alpha_{nl}\right)-y_{l}\left(t-\mathrm{INT}\left(\left(\omega_{w}t\right)/\alpha_{nl}\right)\times \alpha_{nl}/\omega_{w}\right)-\left(r+l\right)\cos\left(\beta \right).$$

According to Equation (43), *h*(*t*) is a periodic function with the period *α_nl_*/*ω_w_*. The value of *h*(*t*) (shown in Figure 7a) for an olig-lugged wheel moving on ST varies as follows during the entire process. When the lug starts to contact the terrain at moment *t*_1_, it decreases until moment *t*_2_ when the lug is parallel to the *y*-axis and reduces to the minimum. Subsequently, *h*(*t*) increases until moment *t*_3_ when the lug departs from the soil and reaches the maximum. Then, it remains constant until the next lug contacts the terrain at moment *t*_4_. For a multi-lugged wheel, there is more than one lug excavating the soil at the same time. The value of *h*(*t*) varies as shown in Figure 7b during the wheel movement. When the lug starts to contact the terrain at moment *t*_1_, *h*(*t*) decreases until moment *t*_2_ when there is one lug in the soil along the *y*-axis direction and reduces to minimum. Subsequently, *h*(*t*) increases until moment ${t^{\prime }}_{3}$ when there are two adjacent lugs symmetrically about the gravity direction and reaches the maximum. Then, it decreases again until the next lug contacts the terrain at moment *t*_4_.

The output displacement of the wheel is
(44)$$y\left(t\right)=q\left(t\right)=h\left(t\right)-\min\left(h\left(t\right)\right).$$

### 2.3. Simplification of Vibration Outputs for Wheels Traversing Different Terrains

The vibration models built in Equations (22), (41), and (44) are too complex to analyze the vibration characteristics, thus they need to be simplified. For ALCT, it is known from Equation (20) that the amounts of time each lug contacts the terrain surface is not equal and is influenced by the waves of the terrain. The amplitude of *y*(*t*) is slightly larger than that for the wheel traversing the reference terrain surface. However, when the wheel rotates 360° on T1 and reference terrain with the same angular velocity, the amount of time and the number of times of the lugs contact the terrain are the same. Therefore, the time of each lug contacting the terrain can be regarded as approximately the same. In addition, *y*(*t*) can be regarded as a periodic function whose amplitude is variable. Compared with the output *y_rt_*(*t*) of the reference vibration, *y*(*t*) can be expressed as follows:(45)$$y\left(t\right)=C_{M}^{1}\left(t\right)\times \max\left(0,(r+l)\cos\left(\varphi_{rt}(t)\right)-r\right)-C_{M}^{1}\left(t\right)\times \max\left(0,\cos\left(\alpha_{nl}/2\right)\left(r+l\right)-r\right).$$
where $C_{M}^{1}(t)$ ($C_{M}^{1}\geq 1$) indicates the increase in the amplitude of *y*(*t*) caused by the terrain roughness compared to the amplitude of *y_rt_*(*t*).

For SLCT, the amplitude of *y*(*t*) is much bigger than *y_rt_*(*t*) for the wheel traversing reference terrain. The number of contact times between wheel and terrain will be less than the number of lugs when the wheel rotates 360°. So, the frequency of *y*(*t*) is lower than *n*_l_*ω*_w_/2π. The output displacement *y*(*t*) of the wheel can be expressed as follows:(46)$$y\left(t\right)=C_{M}^{2}\left(t\right)\times \max\left(0,(r+l)\cos\left(\varphi (t)\right)-r\right)-C_{M}^{2}\left(t\right)\times \max\left(0,\cos\left(\alpha_{nl}/2\right)\left(r+l\right)-r\right)$$
(47)$$\varphi (t)=\theta -\omega_{w}t+\alpha_{nl}\times \mathrm{INT}\left(\left(\omega_{w}t+\alpha_{nl}/2-\theta \right)/\alpha_{nl}\right)\times C_{f}(t).$$
where *C_f_*(*t*) (*C_f_*(*t*) > 1) indicates the frequency reduction of *y*(*t*) incurred by the terrain roughness when compared to the frequency of *y_rt_*(*t*) for the reference terrain. In addition, $C_{M}^{2}(t)$ ($C_{M}^{2}\gg 1$) indicates the increase in the amplitude of *y*(*t*) caused by the terrain roughness compared to the amplitude of *y_rt_*(*t*).

For LEIS, every lug can contact the terrain. According to Equations (43) and (44), the frequency of *y*(*t*) is *n_l_ω_w_*/2π, equal to the frequency of *y_rt_*(*t*). The amplitude of *y*(*t*) is smaller than *y_rt_*(*t*) for the wheel traversing reference terrain. Compared with the output *y_rt_*(*t*) of the reference vibration, *y*(*t*) can be expressed as
(48)$$y\left(t\right)=C_{M}^{3}(t)\times \max\left(0,(r+l)\cos\left(\varphi_{\mathrm{rt}}(t)\right)-r\right)-C_{M}^{3}(t)\times \max\left(0,\cos\left(\alpha_{\mathrm{nl}}/2\right)\left(r+l\right)-r\right).$$
where $C_{M}^{3}(t)$ ($C_{M}^{3}(t)<1$) indicates the decrease in the amplitude of *y*(*t*) incurred from the lug entering the soil when compared to the amplitude of *y_rt_*(*t*).

Equations (45), (46), and (48) show that the amplitude of *y*(*t*) is independent of the angular velocity of the wheel. The value of *y*(*t*), which can be approximated as a periodic function, can be expressed as follows:(49)$$y\left(t\right)=\tilde{y}\left(t\right)+C_{y}=C_{y0}+\sum_{i=1}^{\infty }C_{yi}\cos\left(if_{a}t-\varphi_{i}\right)+C_{y}.$$
where *f_a_* is the frequency of output displacement for the vertical vibration, *φ_i_* is the phase angle of the *i*-th harmonic, and $C_{y}$ is the constant of *y*(*t*) and is equal to −min(*h*(*t*)).

## 3. Analysis and Extraction of Statistical Vibration Features Independent of the Wheel Velocity

It is difficult to measure vertical displacement of a wheel during movement, whereas vertical acceleration of the wheel is easy to obtain. Therefore, the vibration features of the wheels are extracted from the vertical acceleration signal. The vibration acceleration signal is often described by its statistical features such as mean square value, maximum value, kurtosis, and so on. These statistical features are proportional to wheel speed. If a rover moves on low-unevenness terrain at a high speed, the values of statistical features may be close to that for a rover moving on high-unevenness terrain at a low speed. This can cause low-unevenness terrain to be classified as high-unevenness terrain. In order to solve this problem, new speed-independent features are proposed in this paper.

### 3.1. Extraction of Vibration Features Independent of Wheel Velocity

According to the wheel’s vertical vibration displacement *y*(*t*) in Equation (49), its vertical vibration acceleration can be expressed as $a_{y}\left(t\right)=\ddot{y}\left(t\right)=\omega_{w}{}^{2}\left(C_{y}-y\right)=-\omega_{w}{}^{2}\left(C_{y0}+\sum_{i=1}^{\infty }C_{yi}\cos\left(if_{a}t-\varphi_{i}\right)\right)$ (where $C_{y}$ is a constant component of *y*). The vertical vibration acceleration of the wheel can be regard as an approximate period signal, for example, the vertical acceleration of the wheel is an approximate period signal as shown in Figure 8 when the wheel moves on ST. The Fourier-series expansion of a periodic acceleration signal is described as follows:(50)$$a_{y}\left(t\right)=C_{0}+\sum_{i=1}^{\infty }C_{i}\cos\left(if_{a}t-\varphi_{i}\right).$$
where *C_i_* is the amplitude of the *i*-th harmonic.

According to the analysis in Section 2, the changing of wheel–terrain interaction class changes the frequency and amplitude of the vertical vibration acceleration signals of a wheel. The vertical reference acceleration is defined as the vertical acceleration of a wheel when it traverses the reference terrain. Therefore, the vertical reference acceleration can be expressed as $a_{yrt}\left(t\right)={\ddot{y}}_{rt}\left(t\right)=-\omega_{w}{}^{2}\left(C_{yrt0}+\sum_{i=1}^{\infty }C_{yrti}\cos\left(if_{art}t-\varphi_{i}\right)\right)$.

Four statistical features are dimensionlessly processed to be independent of wheel speed. They are as follows:

(1) The frequency ratio of the vertical vibration acceleration for a wheel traversing an actual terrain and the reference terrain is described as
(51)$$\kappa_{f}=f_{a}/f_{art}=\left(2\pi /n_{L}\omega_{w}\right)/\left(2\pi /n_{L}\omega_{w}/{\bar{C}}_{f}\left(t\right)\right)={\bar{C}}_{f}\left(t\right).$$
where *f_a_* is the frequency of the vertical vibration acceleration for wheels traversing actual terrains with angular velocity *ω_w_*, and *f_art_* is the frequency of the vertical vibration acceleration for wheels traversing reference terrain with the same angular velocity *ω_w_*.

The value of *κ_f_* is equal to the average value of *C_f_*(*t*). It is independent of the angular velocity, velocity, and slip ratio of the wheel since *C_f_*(*t*) is independent of those motion parameters. The value of *κ_f_* is directly proportional to the frequency of the wheel in contact with the terrain. According to Section 2, the frequency of the vibration signal is appropriately equal to the quotient of the number of times the wheel contacts the terrain divided by the cost time when the wheel rotates 360°. According to Equations (45), (46), and (48), *κ_f_* is equal to 1 for LEIS and ALCT, and *κ_f_* is smaller than 1 for SLCT.

(2) The ratio of the sum of all harmonic amplitudes in Equation (50) for a wheel traversing an actual terrain and the reference terrain is described as
(52)$$\kappa_{A}=A_{a}/A_{art}=\sum \omega_{w}{}^{2}C_{yi}/\sum \omega_{w}{}^{2}C_{yrti}=\sum C_{yi}/\sum C_{yrti}.$$
where *A_a_* is the sum of all harmonics’ amplitudes of the vertical vibration acceleration for wheels traversing actual terrains with angular velocity *ω_w_*, and *A_art_* is the sum of all harmonics’ amplitudes of the vertical vibration acceleration for wheels traversing reference terrain with the same angular velocity *ω_w_*.

The *κ_A_* is independent of the angular velocity, velocity, and slip ratio of the wheel because $C_{yi}$ and $C_{yrti}$ are independent of those motion parameters. The *κ_A_* is relative to the stiffness and roughness of the terrain. It is directly proportional to the terrain roughness for terrain that does not cause the wheel to sink and inversely proportional to wheel sinkage for terrain that causes the wheel to sink. For ALCT, *κ_A_* is slightly greater than 1. For SLCT, *κ_A_* is much greater than 1. For LEIS, *κ_A_* is less than 1.

(3) The mean value ratio of vertical vibration acceleration for a wheel traversing an actual terrain and the reference terrain is described as follows:(53)$$\kappa_{mv}={\bar{v}}_{a}/{\bar{v}}_{art}=\left(\omega_{w}{}^{2}C_{y0}\right)/\omega_{w}{}^{2}C_{yrt0}=C_{y0}/C_{yrt0}.$$
where ${\bar{v}}_{a}$ is the mean value of the vertical vibration acceleration for wheels traversing actual terrains with angular velocity *ω_w_*, and ${\bar{v}}_{art}$ is the mean value of the vertical vibration acceleration for wheels traversing reference terrain with the same angular velocity *ω_w_*.

Due to $C_{y0}$ and $C_{yrt0}$ being independent of the wheel’s angular velocity, velocity, and slip ratio, the *κ_mv_* is independent of those motion parameters. It indicates intensity of the vertical speed change.

(4) The mean square ratio of the vertical vibration acceleration for a wheel traversing an actual terrain and the reference terrain is described as follows:(54)$$\kappa_{ms}={\bar{s}}_{a}/{\bar{s}}_{art}=\left(\bar{\omega_{w}{}^{4}a_{y}{}^{2}\left(t\right)/\omega_{w}{}^{4}}\right)/\left(\bar{\omega_{w}{}^{4}a_{rty}{}^{2}\left(t\right)/\omega_{w}{}^{4}}\right)=\left(\bar{a_{y}{}^{2}\left(t\right)/\omega_{w}{}^{4}}\right)/\left(\bar{a_{rty}{}^{2}\left(t\right)/\omega_{w}{}^{4}}\right).$$
where ${\bar{v}}_{a}$ is the mean square value of the vertical vibration acceleration for wheels traversing actual terrains with angular velocity *ω_w_*, and ${\bar{v}}_{art}$ is the mean square value of the vertical vibration acceleration for wheels traversing reference terrain with the same angular velocity *ω_w_*.

The value of *κ_ms_* is independent of the wheel’s angular velocity, velocity, and slip ratio because the numerical value of $a_{y}{}^{2}\left(t\right)/\omega_{w}{}^{4}$ is independent of those motion parameters. The *κ_ms_* is relative to the stiffness and roughness of the terrain, similar to *κ_A_*. It can reflect the variation in speed of the signal from the minimum to the maximum value. It is the powerfulness of vibration energy. For ALCT, *κ_ms_* is slightly greater than 1. For SLCT, *κ_ms_* is much greater than 1. For LEIS, *κ_ms_* is smaller than 1.

### 3.2. Analysis of Vibration Features Based on a Single-Wheel Experiment

Experiments were conducted using the wheel–terrain interaction test platform with an accelerometer used to collect the vertical acceleration of a wheel, as shown in Figure 9a. The three motors on the platform are the driving, turning, and dragging motors, and the sensors on the platform include the torque, displacement, and six-axis force/torque sensor. The driving motor drives the wheel forward, and the dragging motor is used to simulate the body movement of the rover. A wheel moving with different slip ratios can be controlled by regulating the two motors at different rotational speeds. The displacement sensor is used to measure wheel sinkage. A counterweight is used to adjust vertical load of the wheel, whereas the actual load is measured using the six-axis force/torque sensor. The radius of the wheel is 140 mm and its width is 150 mm. A wheel with 28 lugs of 15 mm in height is shown in Figure 9b. The vertical load of the wheel is 100 N. For each terrain class, the velocities of the wheel are 10, 14, and 20 mm/s, and the slip ratio of the wheel is 0–0.4.

The Mars terrain mainly includes hard, sandy, and gravel surfaces [3]. Rovers need to safely traverse those terrains. Motivated by the challenges of rover mobility, HT, GT, ST, hard terrain with gravel (HGT), and sandy terrain with gravel (SGT) were chosen for the experiment, as indicated in Figure 10. HT is made up of hard slate, GT is constructed with a pile of gravel, ST is sand, HGT is formed by placing gravel randomly on slate, and SGT is sand with a small amount of gravel with a random distribution.

The wheel–HT interaction pertains to ALCT, wheel–GT/HGT interaction pertains to SLCT, and wheel–ST/SGT interaction belongs to LEIS. Figure 11 shows the extraction results for the vibration features. The values of the vibration features are expressed as a fuzzy set {*Small* (*S*), *Medium* (*M*), *Big* (*B*), *Very-Big* (*VB*)}. Table 2 shows a comparison of all vibration features for three WTICs. The vibration features extracted to constitute feature vector ***P*** for WTIC recognition are {*p*_1_, *p*_2_, *p*_3_, *p*_4_} (*p*_1_ = lg(*κ_f_*), *p*_2_ = lg(*κ_A_*), *p*_3_ = lg(*κ_mv_*), and *p*_4_ = lg(*κ_ms_*)) According to ***P***, each WTIC can be distinguished from the others.

## 4. Recognition of Wheel–Terrain Interaction Class Based on Speed-Independent Vibration Features

Every WTIC has a suitable terramechanics model. Based on WTIC recognition, the terramechanics model can be correspondingly selected for identifying terrain parameters. A vibration-based speed-independent WTIC recognition method without requirement of training data is introduced in this section.

### 4.1. Analysis of Feature Space of the Four Speed-Independent Features

It can be known from Table 2 that the WTIC can be distinguished according to its vibration features’ values. The regions of feature space formed by the vibration features for three WTICs can be partitioned as shown in Figure 12. The value ranges of feature vector ***P*** for three WTICs can be known from the analysis of Section 3.2. When the WTIC is ALCT, *κ_f_* is approximately equal to 1, and *κ_A_* and *κ_ms_* are slightly bigger than 1. Thus, *p*_1_ is approximately equal to 0, and *p*_2_ and *p*_4_ are slightly bigger than 0. As every lug is in contact with the terrain, there are upper limits for *p*_2_ and *p*_4_. The experiment results show that *p*_3_ is also slightly greater than zero. The feature vector ***P*** for ALCT (such as wheel–HT interaction) is located within the vicinity of zero. The vicinity of zero is expressed as {*p*_1_, *p*_2_, *p*_3_, *p*_4_|lg*ε_fl_* < *p*_1_ < lg*ε_fu_*, lg*ε_Al_* < *p*_2_ <lg*ε_Au_*, lg*ε_mvl_* < *p*_3_ <lg*ε_mvu_*, lg*ε_msl_* < *p*_4_ < lg*ε_msu_*}, where *ε_fl_*, *ε_Al_*, *ε_mvl_*, and *ε_msl_* are the lower limits for *κ_f_*, *κ_A_*, *κ_mv_*, and *κ_ms_* respectively, and *ε_fu_*, *ε_Au_*, *ε_mvu_*, and *ε_msu_* are the upper limits for *κ_f_*, *κ_A_*, *κ_mv_*, and *κ_ms_*, respectively.

According to the analysis of Section 3.2, when the WTIC is SLCT, *κ_f_* is smaller than 1, and *κ_A_* and *κ_ms_* are greater than 1. The feature vector ***P*** for SLCT (such as wheel–HGT/GT interaction) is located in the region {*p*_1_, *p*_2_, *p*_3_, *p*_4_ | *p*_1_ < lg*ε_fl_*, lg*ε_Au_* < *p*_2_, lg*ε_mvu_* < *p*_3_, lg*ε_msu_* < *p*_4_}. When WTIC is LEIS, *κ_f_* is approximately equal to 1, and *κ_A_*/*κ_ms_* are smaller than 1. The feature vector ***P*** for LEIS (such as wheel–ST/SGT interaction) is in the region {*p*_1_, *p*_2_, *p*_3_, *p*_4_ | lg*ε_fl_* < *p*_1_ < lg*ε_fu_*, *p*_2_ < lg*ε_Al_*, lg*ε_mvl_* < *p*_3_ <lg*ε_mvu_*, *p*_4_ < lg*ε_msl_*}. If ***P*** is not located in the above three regions, the corresponding WTIC is classified as an unknown wheel–terrain interaction class (UWTIC).

### 4.2. Analysis of Feature Vector **P** for the All Lugs Contacting Terrain Surface within the Vicinity of Zero

Owing to the wheels used on the rover all being multi-lugged, the analysis in this section is conducted without considering olig-lugged wheels. When the WTIC is ALCT, the supporting lug of the wheel must depart from the terrain before the adjacent lug along the *y*-axis. When lug lu_1_ contacts the terrain after lug lu_0_ along the *y*-axis and departs from the terrain before the adjacent lug lu_2_ along the *y*-axis, as shown in Figure 13, the vibration displacement caused by lu_1_ has the highest amplitude in this case. When lu_1_ is along the *y*-axis, the distance *h*(*t*) between the lowest point of the wheel and the terrain is *l*. The distance *h*(*t*) calculated using lu_1_ is $\left(r+l\right)\cos\left(\alpha_{\mathrm{nl}}\right)-r$ at the moment when lug lu_2_ is along the *y*-axis. The amplitude of the vibration output displacement caused by rotating around lu_1_ is $\left(r+l\right)\left(1-\cos\left(\alpha_{\mathrm{nl}}\right)\right)$. Thus, to make every lug have contact with the terrain, the amplitude of the vibration output displacement caused by rotating around each lug is no bigger than $\left(r+l\right)\left(1-\cos\left(\alpha_{\mathrm{nl}}\right)\right)$. The amplitude of the reference vibration displacement is $\left(r+l\right)\left(1-\cos\left(\alpha_{\mathrm{nl}}/2\right)\right)$. Therefore, $C_{M}^{1}\left(t\right)$ in Equation (45) is no bigger than $\left(1-\cos\left(\alpha_{\mathrm{nl}}\right)\right)/\left(1-\cos\left(\alpha_{\mathrm{nl}}/2\right)\right)$. The following in equalities can be obtained based on Equations (52) and (54).
(55)$$\kappa_{A}\leq \left(1-\cos\left(\alpha_{\mathrm{nl}}\right)\right)/\left(1-\cos\left(\alpha_{\mathrm{nl}}/2\right)\right).$$
(56)$$\kappa_{ms}\leq {{\left(1-\cos\left(\alpha_{\mathrm{nl}}\right)\right)}^{2}/\left(1-\cos\left(\alpha_{\mathrm{nl}}/2\right)\right)}^{2}.$$

Thus, the following equations can be obtained:(57)$$\varepsilon_{Au}=\left(1-\cos\left(\alpha_{\mathrm{nl}}\right)\right)/\left(1-\cos\left(\alpha_{\mathrm{nl}}/2\right)\right).$$
(58)$$\varepsilon_{msu}={{\left(1-\cos\left(\alpha_{\mathrm{nl}}\right)\right)}^{2}/\left(1-\cos\left(\alpha_{\mathrm{nl}}/2\right)\right)}^{2}.$$

The parameters *ε_fl_*, *ε_mvl_*, *ε_fu_*, and *ε_mvu_* are influenced by the noise and accelerometer properties including the sampling frequency and precision. In this study, *ε_fl_* = 1 − *ε_f_*, *ε_fu_* = 1 + *ε_f_*, *ε_f_* is set as 0.15, *ε_mvu_* is set as 10, and the lower limits of *ε_Al_*, *ε_mvl_*, and *ε_msl_* are 0.5. Therefore, the range of the feature vector ***P*** for the ALCT is as follows:(59)$$\left\{\begin{array}{l} \mathrm{lg}\left(1-\varepsilon_{f}\right)\leq p_{1}\leq \mathrm{lg}\left(1+\varepsilon_{f}\right)\\ \mathrm{lg}\varepsilon_{Al}\leq p_{2}\leq \mathrm{lg}\left(\left(1-\cos\left(\alpha_{\mathrm{nl}}\right)\right)/\left(1-\cos\left(\alpha_{\mathrm{nl}}/2\right)\right)\right)\\ \mathrm{lg}\varepsilon_{mvl}\leq p_{3}\leq \mathrm{lg}\left(\varepsilon_{mvu}\right)\\ \mathrm{lg}\varepsilon_{msl}\leq p_{4}\leq \mathrm{lg}\left({{\left(1-\cos\left(\alpha_{\mathrm{nl}}\right)\right)}^{2}/\left(1-\cos\left(\alpha_{\mathrm{nl}}/2\right)\right)}^{2}\right) \end{array}\right..$$

### 4.3. Recognition of Wheel–Terrain Interaction Class Based on Vibration Features

According to the value of feature vector ***P***, the WTIC can be recognized. If ***P*** satisfies Equation (59), the WTIC is ALCT. According to the analysis in Section 4.1, the value ranges of feature vector ***P*** for SLCT and LEIS are expressed as Equation (60) and Equation (61), respectively.
(60)$$\left\{\begin{array}{l} p_{1}<\mathrm{lg}\left(1-\varepsilon_{f}\right)\\ \mathrm{lg}\left(\left(1-\cos\left(\alpha_{\mathrm{nl}}\right)\right)/\left(1-\cos\left(\alpha_{\mathrm{nl}}/2\right)\right)\right)<p_{2}\\ \mathrm{lg}\varepsilon_{mvu}\leq p_{3}\\ \mathrm{lg}\left({{\left(1-\cos\left(\alpha_{\mathrm{nl}}\right)\right)}^{2}/\left(1-\cos\left(\alpha_{\mathrm{nl}}/2\right)\right)}^{2}\right)<p_{4} \end{array}\right..$$
(61)$$\left\{\begin{array}{l} \mathrm{lg}\left(1-\varepsilon_{f}\right)\leq p_{1}\leq \mathrm{lg}\left(1+\varepsilon_{f}\right)\\ p_{2}<\mathrm{lg}\varepsilon_{Al}\\ \mathrm{lg}\varepsilon_{mvl}\leq p_{3}\leq \mathrm{lg}\left(\varepsilon_{mvu}\right)\\ p_{4}<\mathrm{lg}\varepsilon_{msl} \end{array}\right..$$

If ***P*** is not located within the above three regions, the corresponding WTIC is UWTIC. The programming for WTIC recognition is shown in Figure 14. Table 3 shows the WTIC recognition results for the single-wheel test experiments.

The accuracy of the WTIC categorization into the three classes is no less than 96.67%. These results show the ability of the algorithm to distinguish WTICs. When attempting to identify ALCT, the algorithm misclassifies 2 samples of 60 samples as UWTIC, and the recognition accuracy reaches 96.67%. When recognizing SLCT, 2 samples of 120 samples are misclassified as UWTIC, and the recognition accuracy is about 98.34%. Similarly, when recognizing LEIS, 4 samples of 120 samples are misclassified as UWTIC, and the recognition accuracy is approximately 96.67%. The three WTICs are not mutually misclassified. This clearly demonstrates the ability of the algorithm to identify WTICs that induce clearly distinct vibrations on the wheel.

## 5. Estimation of Terrain Properties Based on the Recognition of Wheel–Terrain Interaction Class

The WTICs can be recognized using the algorithm shown in Figure 14. The terramechanics model is switched corresponding to the recognized WTIC and then used to identify the terrain parameters in this section.

### 5.1. Switchable Terramechanics Model

Yang et al. [19] inferred that the terramechanics models for wheels moving on sand and hard terrain can be similarly expressed, and the parameters of sand and hard terrain can be unified. The terramechanics model equations for wheel–ST interaction can be switched to the equations that have the same form as the Hunt–Crossley contact model by setting the shearing deformation modulus *K* and soil cohesion *c* as zero. It can be used for the interaction between a rigid wheel and hard terrain.

The normal force *F_N_*, drawbar pull *F_DP_*, and resistance moment *M*_R_ between the wheel and soil can be expressed as follows:(62)$$F_{N}=rb\left(A+BE\tan\varphi \right)k_{s}z^{N}+rbcBE.$$
(63)$$F_{DP}=rb\left(AE\tan\varphi -B\right)k_{s}z^{N}+rbcAE.$$
(64)$$M_{R}=r^{2}bcCE+r^{2}bCE\tan\varphi k_{s}z^{N}.$$
where *r* is the wheel radius; *b* is the wheel width; *z* is the wheel sinkage, i.e., deformation of the terrain; *c* is the soil cohesion; *φ* is the internal friction angle of the soil; *N* is the sinkage exponent (*N* = *n*_0_ + *n*_1_*s*); *k_s_* is expressed as *k_s_ = k_c_*/*b + k_φ_* (where *k_c_* is the cohesive modulus of the soil and *k_φ_* is frictional modulus of soil); and *A*, *B*, and *C* are as follows:(65)$$\left\{\begin{array}{l} A=\frac{\cos\theta_{m}-\cos\theta_{2}}{\theta_{m}-\theta_{2}}+\frac{\cos\theta_{m}-\cos\theta_{1}}{\theta_{1}-\theta_{m}}\\ B=\frac{\sin\theta_{m}-\sin\theta_{2}}{\theta_{m}-\theta_{2}}+\frac{\sin\theta_{m}-\sin\theta_{1}}{\theta_{1}-\theta_{m}}\\ C=\frac{\theta_{1}-\theta_{2}}{2} \end{array}\right..$$
where *θ*_1_ (*θ*_1_ = arccos(1-*z*/*r*)) is the entrance angle, *θ_m_* (*θ_m_* = (*c*_1_ + *c*_2_*s*)*θ*_1_) is the angle of the maximum stress, *θ*_2_ (*θ_2_* = *c*_3_*θ*_1_) is the departure angle, and *c*_1_, *c*_2_, and *c*_3_ are the coefficients of wheel–soil interaction angle. The parameter *E* is expressed as follows:(66)$$E=1-\exp(-j(\theta_{m})/K).$$
where *K* represents the shearing deformation modulus, and *j* is the shearing displacement. When the wheel contacts hard terrain, the terrain has no shearing deformation. Thus, *K* is equal to zero. In addition, *c* can be set to zero according to Yang et al. [19]. Equations (62)–(64) are as follows:(67)$$F_{N}=rb\left(A+B\tan\varphi \right)k_{s}z^{N}=kz^{N}.$$
(68)$$F_{DP}=\frac{A\tan\varphi -B}{\left(A+B\tan\varphi \right)}F_{N}=\mu F_{N}.$$
(69)$$M_{R}=r\frac{C\tan\varphi }{A\tan\varphi -B}F_{DP}=\lambda_{c}rF_{DP}.$$

For interaction between the wheel and hard terrain, *θ*_1_ tends toward zero, and thus the coefficient of friction *μ* tends toward tan*φ*. The parameter *λ_c_* is the coefficient used to balance the driving torque. When the parameters *K* and *c* are appropriately assigned, the terramechanics model can be used for a rover traversing hard terrain. Therefore, the terramechanics model for wheel–ST interaction can be switched to be suitable for wheel–HT interaction.

### 5.2. Unified Identification Model for Terrain Parameters

Let X=tanφE, $Y=rbk_{s}z^{N}$, and Q=rbcE; the following equation can be obtained:(70)$$\left[\begin{array}{c} F_{N}\\ F_{DP}\\ M_{R} \end{array}\right]=\left[\begin{array}{cccc} A & 0 & B & B\\ 0 & -B & A & A\\ 0 & 0 & rC & rC \end{array}\right]\left[\begin{array}{c} X\\ Y\\ XY\\ Q \end{array}\right].$$

Therefore,
(71)$$XY+Q=M_{R}/\left(rC\right).$$
(72)$$Y=\left(rCF_{N}-BM_{R}\right)/\left(rCA\right).$$
(73)$$F_{DP}=AY+B(XY+Q)=\frac{A^{2}+B^{2}}{AC}\frac{M_{R}}{r}-\frac{B}{A}F_{N}.$$
(74)$$W=F_{N}=AY+B(XY+Q)=Arbk_{s}z^{N}+\frac{B}{C}\frac{M_{R}}{r}.$$
(75)$$T_{D}=M_{R}=r^{2}CE\left[bc+F_{N}\tan\varphi /\left(rA\right)\right]/\left[1+BE\tan\varphi /A\right].$$
where *W* and *T_D_* are the wheel load and driving torque of the motor, respectively. Equations (73)–(75) are a decoupled terramechanics model regarding unknown terrain parameters *K_s_*, *n*_0_, *n*_1_, *c*_1_, *c*_2_, *c*, *φ*, and *K*, which can be used for terrain parameter identification.

### 5.3. Identification of Terrain Parameters Based on Terramechanics Model Switching

Table 4 shows the correspondence between WTICs and setting values of terrain parameters for terramechanics model switching. WTIC recognition is used to judge whether some parameters should be set to zero to achieve terramechanics model switching. If the WTIC is ALCT, *c* is set to zero, *K* is set to zero, *E* is set to 1. This is because there is no shear deformation or backward accumulation of the soil for ALCT. GT is a typical terrain pertaining to T2 (wheel–T2 interaction is SLCT). When the wheel traverses GT, gravel will move slightly backward and accumulate, which is far less than the accumulation degree of sand. Thus, if the WTIC is SLCT, according to [19], *c* is set to zero.

The sinkage and slip ratio of the wheel are necessary parameters for identification of the terrain parameters in the terramechanics model. Wheel sinkage can be detected using machine vision methods or through a kinematic analysis of the rover suspension [21,22]. When WTIC is not LEIS, sinkage is set as 0.01mm for terrain parameter identification. The wheel slip ratio can be estimated using a machine vision method [23]. The vertical load *W*, driving torque *T_D_*, and drawbar pull *F_DP_* can be measured by installing a six-axis force/torque sensor on the wheel [18].

According to Equation (73), *F_DP_* is a function of *W*, *T_D_*, *θ*_1_, *s*, and the unknown parameters *c*_1_ and *c*_2_. Here, *c*_1_ and *c*_2_ can be identified according to the measured *F_DP_*, *W*, *T_D_*, *θ*_1_, and *s*. According to Equation (74), *W* is a function of *T_D_*, *θ*_1_, *z*, *s*, identified parameters *c*_1_ and *c*_2_, and unknown parameters *K_s_*, *n*_0_, and *n*_1_. The three parameters *K_s_*, *n*_0_, and *n*_1_ can be identified using the measured *W*, *T_D_*, *θ*_1_, *z*, *s*, and identified *c*_1_, *c*_2_. According to Equation (75), *T_D_* is a function of *W*, *θ*_1_, *s*, identified parameters *c*_1_ and *c*_2_, and unknown parameters *c*, *φ*, and *K*. Parameters *c*, *φ*, and *K* can be identified using the measured *W*, *T_D_*, *θ*_1_, *s*, and identified values of *c*_1_ and *c*_2_.

The least squares method is used to achieve identification for unknown terrain parameters. When identifying *c*_1_ and *c*_2_, *xe* denotes the parameter vector {*c*_1_, *c*_1_} to be identified; *xdata* denotes the input data vector {*s*, *F_N_*, *T_D_*}; *ydata* denotes the measured *F_DP_*; *m* denotes the length of *xdata* and *ydata*; and *F* denotes the function of Equation (73). When identifying *K_s_*, *n*_0_, and *n*_1_, *xe* denotes the parameter vector {*K_s_*, *n*_0_, *n*_1_} to be identified; *xdata* denotes the input data vector {*s*, *T_D_*}; *ydata* denotes the measured *F_N_*; and *F* denotes the function of Equation (74). When identifying *c*, *φ*, and *K*, *xe* denotes the parameter vector {*c*, *φ*, *K*} to be identified; *xdata* denotes the input data vector {*s*, *F_N_*}; *ydata* denotes the measured *T_D_*; and *F* denotes the function of Equation (75). The ultimate goal is to find the vector *xe* that best fits Equation (76):(76)$$\underset{xe}{\min}\frac{1}{2}\left\|F(xe,xdata)-ydata\right\|=\frac{1}{2}\sum_{i=1}^{m}{\left[F(xe,xdata_{i})-ydata_{i}\right]}^{2}.$$

Figure 15 shows the programming of the terrain parameter identification for different types of terrain.

## 6. Experiments for Wheel–Terrain Interaction Class Recognition and Terrain Parameter Identification Using a Planetary Rover Prototype

The method proposed in this paper is experimentally validated using a six-wheeled Planetary Rover Prototype. The collected vibration acceleration of the wheel is used to recognize the WTIC. WTIC recognition guides the rover to switch the terramechanics model for identifying terrain parameters in the model using the algorithm shown in Figure 15. The results for the WTIC recognition and identification of terrain parameters in the terramechanics model are discussed in this section.

### 6.1. Introduction of Planetary Rover Prototype and Terrain Environment

The Planetary Rover Prototype is shown in Figure 16. An accelerator mounted on the front left leg of the rover, near the joint with the wheel axle, is used to collect the vertical acceleration of the wheel to study the vibration-based WTIC recognition. A camera is used to monitor the robot’s wheel and detect the wheel sinkage and slip ratio. The camera is mounted on a supporting arm, with three moving degrees of freedom (DOFs) and three rotational DOFs, installed on the robot body. A gyroscope is attached to the camera to detect the rotation angle of the camera. The position of the camera can be acquired with the relation of the arm and robot body. Every wheel is equipped with a six-axis force/torque sensor to detect the drawbar pull force *F_DP_*, vertical load *W* (equal to normal force *F_N_*), lateral force *F_L_*, driving moment *T_D_*, steering moment *M_S_*, and overturning moment *M_O_*. The radius of the wheel is 155 mm and its width is 150 mm. The wheel has 24 lugs of 10 mm in height.

Three different surfaces were selected for the experiments: GT, HT (tiled with stone bricks), and ST, as shown in Figure 17. HT pertains to terrain T1, GT pertains to terrain T2, and ST pertains to terrain T3. As such, they represent a wide range of hardnesses from soft to hard, and a roughness ranging from flat to rugged, thus better illustrating the overall accuracy of the methods presented.

### 6.2. Recognition of Wheel–Terrain Interaction Class for Planetary Rover Prototype

On each of the terrain surfaces, the rover drove back and forth thrice at a constant velocity of 0.003, 0.004, 0.005, 0.006, 0.007, 0.008, 0.009, and 0.01 m/s. During the rover movement, 32-bit vibration acceleration samples were collected using the accelerator shown in Figure 14 at a frequency of 128 kHz. The signal is processed by down-sampling and low-pass filtering to extract vibration features. Figure 18 and Figure 19 show the vibration signal samples for three classes of terrain. Once the single-terrain datasets were collected, multi-terrain datasets were collected with the rover driving from ST to HT to GT in a single traversal, both forward and reverse.

A single dataset consists of vibration data from an about 55 s traversal period. The acceleration signals are segmented with 15 s long intervals to recognize WTIC. Table 5 show the results for the WTIC recognition of single-terrain datasets. They demonstrate an excellent performance of the WTIC recognition method proposed in this paper in distinguishing the WTICs. The results show that 133 samples of 144 wheel–HT interaction (which belongs to ALCT) are recognized as ALCT, and the recognition accuracy reaches 92.36%. Ten samples of wheel–GT interaction (which belongs to SLCT) are misclassified as UWTIC, i.e., 93.06% of the wheel–GT interactions are classified as SLCT. All wheel–ST interaction samples are recognized as LEIS, and the accuracy attains 100%. No misidentification occurred among the three WTICs. The results show that the WTIC recognition algorithm has a high level of accuracy and can effectively distinguish the WTICs.

Figure 20 shows information regarding the vibration acceleration and WTIC recognition results for a distinct single traversal of all three terrain classes shown in Figure 15. The rover is traveling from an area of ST to an area of HT and then on to an area of GT. The data are also divided into 15 s interval segments for WTIC recognition. During the entire movement, the Planetary Rover Prototype operates for a total of 315 s, thus, the acceleration is divided into 21 segments for WTIC recognition. The segmentation is shown in the dotted square in Figure 18, and the WTIC recognition results are represented by the mark at the bottom of the block. These results clearly show the effectiveness of the algorithm in recognizing multiple WTICs during a single traversal. This figure also illustrates the most common error mechanism of the algorithm, which occurs when the rover transitions between homogenous terrain regions.

### 6.3. Terrain Parameter Identification for Planetary Rover Prototype

The wheel sinkage and slip ratio are necessary parameters for terrain parameter identification and can be detected using machine vision methods shown in our earlier studies [22,23].

#### 6.3.1. Introduction of Sinkage Detection for Wheels Moving on Soft Terrain

Ref. [22] detailed the detection method of wheel sinkage, as shown in Figure 21.

It proposes a new definition of wheel sinkage for rough terrain. Wheel sinkage has been defined as a vector using the wheel–soil boundary. A new vision-based method for sinkage detection is proposed in the paper. The saturation of wheel–terrain image is adjusted using a dynamic piecewise non-linear adjustment to enhance the color contrast between the wheel and soil regions. Through a binarizing process, the wheel section in the color-enhanced image is segmented to acquire an image in which the wheel region is white and other regions are black. Thus, the outline of the wheel region can be extracted from the binary image using the edge detection method. The wheel section outline is corrected to represent the wheel rim operating as a circle. The distance between points of the wheel rim and wheel center is equal to the wheel radius. And the distance between points of the wheel–soil boundary and wheel center is less than the wheel radius. The wheel–soil boundary is extracted based on this characteristic. The sinkage is calculated using the wheel–soil boundary based on the sinkage definition.

#### 6.3.2. Introduction of Wheel Slip Ratio Detection

Ref. [23] detailed a detection method for wheel slip ratio shown in Figure 22. The wheel slip ratio was estimated using the same wheel–terrain images as used in the sinkage detection. At different moments, the wheel position in the wheel–terrain image is the same, while the position of the wheel–trace boundary is different. Thus, the wheel speed can be estimated from the displacement of the wheel trace relative to the wheel at different moments. Based on this idea, Lv et al. [23] built an estimation model for the slip ratio, which is expressed as:(77)$$s=1-\frac{\Delta S}{\arccos\left(1-\Delta u^{2}/2r_{1}{}^{2}\right)r}$$
where Δ*S* is the displacement of the wheel, Δ*u* is the average displacement of the mark points, and *r*_1_ is the radius of the points.

As shown in Figure 22a, the boundaries of the wheel trace were extracted from wheel–terrain images acquired at adjacent moments and were matched to estimate the forward displacement Δ*S* of the wheel. Meanwhile, the mark points (in this paper, the screws on the wheel are regarded as mark points) of the wheel were also extracted from wheel–terrain images acquired at adjacent moments and were matched to estimate the average displacement Δ*u* of the mark points relative to the wheel center. The slip ratio can then be calculated using Equation (77) according to Δ*S* and Δ*u*.

When the rover is traversing on HT or GT, there is no obvious wheel trace on the terrain, but there are SURFF feature points that do not exist on ST. The wheel trace boundary extraction and matching proposed by Lv et al. [23] for slip ratio estimation are replaced by the terrain SURFF feature point extraction and matching. The forward displacement Δ*S* of the wheel can be estimated by detecting the SURFF feature points in the wheel–terrain images acquired at two adjacent moments, thereby estimating the wheel slip ratio, as shown in Figure 22b.

#### 6.3.3. Identification Results of Terrain Parameters for Three Classes of Terrain

The slip ratio of the object wheel can be adjusted by varying the speed of the object wheel and maintaining other wheels at a constant speed. For each terrain class, the ratio is controlled at 0, 0.05, 0.1, 0.15, 0.2, 0.3, 0.4, 0.5, and 0.6. The actual slip ratio is detected visually.

Terrain parameters can be divided into three groups, namely, ***P***_I_ = {*c*_1_, *c*_2_}, ***P***_II_ = {*k_s_*, *n*_0_, *n*_1_}, and ***P***_III_ = {*c*, *φ*, *K*}, which are directly related to angles of *θ*_m_ and *θ*_2_, normal stress, and shearing stress, respectively. The results of terrain parameter identification using the algorithm in Figure 15 are shown in Table 6. Figure 23 and Figure 24 show comparison of the parameters ***P***_II_ = {*k_s_*, *n*_0_, *n*_1_} and ***P***_III_ = {*c*, *φ*, *K*} for three terrain classes, respectively, to clearly illustrate the difference in terrain parameters between the three terrain classes.

Wheel–terrain interaction forces are predicted for the wheels without considering the rover dynamics or force coordination among the wheels during the rover movement to simplify the calculation of them for easy control. The estimated drawbar pull force *F_DP_*, wheel load *W*, and driving torque *T_D_* with identified terrain parameters are compared with the sensor-measured results as shown in Figure 25, Figure 26 and Figure 27. Their estimated relative errors are shown in Table 7. From Figure 25, Figure 26 and Figure 27 and Table 7, the predicted wheel–terrain interaction forces are close to the measured values. Thus, the simplified predicted wheel–terrain force can be used for optimal control (such as to maximize the wheel traction or minimize the power consumption) for rovers.

#### 6.3.4. Analysis of Parameter Identification Results of Hard Terrain

Table 6 shows that the *k_s_* of HT is much larger than that of ST because the internal bonding force of HT is much greater than that of ST and has a large rigidity; thus, the terrain does not undergo substantial deformation when a wheel moves on it. There is no sinkage of the wheel. The sinkage exponents *n*_0_ and *n*_1_ have different characteristics for three classes of terrain. The *n*_0_ of HT is close to 0.5, whereas *n*_1_ is approximately zero. Thus, the sinkage exponent *N* is not affected by wheel slip ratio. The normal force model of the wheel–HT interaction is a non-linear elastic model as shown in Equation (67). The tangential bond of HT is extremely large. During the interaction between the wheel and HT, the soil will not be damaged. There is no shear deformation or backward accumulation for the terrain. Therefore, there is no shear stress in the force of the wheel–terrain interaction, and only normal stress occurs. From Figure 25a, the fitting curve of estimated wheel load *W* using the identified parameters is extremely close to the fitting curve of the measured data. The relative error of estimated wheel load *W* is no more than 11.68% as shown in Table 7.

The drawbar pull force can be calculated using the friction model as shown in Equation (68). When the wheel slips, the drawbar pull force of the wheel is equal to the friction between the wheel and the terrain. The parameters *c* and *K* are both zero, *φ* represents the external friction angle of the terrain, and tan*φ* is the friction coefficient between the terrain and wheel.

Because the wheel sinkage is close to zero, the maximum stress angle *θ_m_* is calculated as equal to zero according to the coefficients of the wheel–soil interaction angle.

Figure 25a shows that the estimated values of *F_DP_* and *T_D_* differ greatly from the measured values when slip ratio is 0. And the maximum relative error of the estimated drawbar pull force is 65.8%, as shown in Table 7. However, the estimated *F_DP_* and *T_D_* are close to measured data except when the slip ratio is 0, as shown in Figure 25b. The relative errors of estimated drawbar pull force and driving torque in Figure 25b are both no more than 15.52%. When the slip ratio is zero, there is no relative movement between the wheel and terrain, and the drawbar pull force (friction force) of the wheel is static friction force. At this time, the actual drawbar pull force (friction force) is smaller than the dynamic friction force calculated according to *φ* (the friction coefficient between wheel and terrain is tan(*φ*)). Thus, the errors of the estimated drawbar pull force and the driving torque are large when slip ratio is zero. Figure 25b shows a comparison among the estimated values of the drawbar pull force, the normal load, and driving torque and the sensor measurement values when the slip ratio is bigger than zero. The fitting curve of estimated data is close to the fitting curve of measured data. The relative errors of estimated wheel–HT interaction force with identified terrain parameters are less than 16% when slip ratio is not zero.

#### 6.3.5. Analysis of Parameter Identification Results of Gravel Terrain

Table 6 shows that GT’s *k_s_* is close to HT’s, GT has a large rigidity, and the terrain does not undergo substantial deformation when a wheel moves on it. GT’s *n*_0_ is close to 0.5 and *n*_1_ is about 0.054. The value of *n*_1_ is relatively small, but the value of *k_s_* is very large. The normal force is sensitive to the sinkage exponent. Therefore, the normal force in Figure 26 has obvious variation.

Under the action of wheels, gravel will slightly move backward and accumulate, which is far less than the accumulation degree of sand. There is shear displacement between pieces of gravel and shear stress in the force of the terrain on the wheel. Therefore, *K* is not 0 in the shear parameter, and *φ* represents the external friction angle of the terrain.

Since the wheel sinkage is close to 0, the maximum stress angle calculated from coefficients of the wheel–soil interaction angle is approximately equal to 0.

The drawbar pull force, normal load, and driving torque of the wheel are estimated using Equations (62)–(64) with the identified parameters. Figure 26 shows that the fitting curves of estimated wheel interaction force are close to the fitting curves of measured data. The maximum errors of estimated drawbar pull force, wheel load, and driving torque are 11.85%, 6.67%, and 8.46% respectively.

#### 6.3.6. Analysis for Parameter Identification Results of Sandy Terrain

Table 6 shows that *k_s_* of ST is much smaller than that of GT/HT. The bonding force between sand grains is small, and thus the stiffness of sand is low. The wheel easily sinks into the soil when the rover traverses over ST. The *n*_0_ of ST is about 0.8169 and *n*_1_ is about 1.2957. Thus, the sinkage exponent *N* is greatly affected by the wheel slip ratio. The normal stress model is a variable exponential non-linear elastic model as shown in Equation (62).

During the interaction between the wheel and terrain, the soil will be displaced, causing the compression under the wheel and accumulation behind the wheel. The terrain places a large amount of shear stress on the wheel, which is not symmetrically distributed along the *y*-direction of the wheel. The shear stress is also an important factor in calculating the supporting force of the terrain applied to the wheel.

Owing to the obvious shear deformation, both *c* and *K* are greater than zero and much larger than those of GT. In addition, *φ* is the internal friction angle of the soil.

From Figure 27, it can be seen that the relative error of the estimated drawbar pull force, wheel load, and driving torque using Equations (62)–(64) with the identified parameters is small, and the fitting curves of the estimated data are extremely close to the fitting curves of the measured data. Table 7 shows that the maximum relative errors of the estimated drawbar pull force, wheel load, and driving torque are 8.11%, 7.64%, and 7.75%, respectively.

According to the above analysis of experiment results, it can be seen that the relative errors of estimated wheel–terrain interaction force with identified terrain parameters are less than 16%, 12%, and 9% for rovers traversing hard, gravel, and sandy terrain, respectively.

## 7. Discussion and Conclusions

This paper has been focused on speed-independent vibration-based WTIC recognition. The terramechanics model was switched for terrain parameter identification according to the recognized WTIC. Concluding remarks of this study are provided as follows:

(1) Three vibration models of rigid lugged wheels have been built and analyzed for three WTICs. The three vertical vibration displacements are all approximately periodic functions. Therefore, the vibration acceleration of a rigid lugged wheel was approximated as a periodic signal. Its frequency, amplitude, mean value, and mean square value were extracted as vibration features describing the wheel–terrain interaction response characteristics and processed to make them non-dimensional and independent of the wheel speed.

(2) A vibration-based method for WTIC recognition without requirement of training data is proposed. The regions of the vibration feature space for three WTICs are analyzed. The WTICs are determined by judging which feature space region the wheel–terrain interaction is located in. Experiment results of the single-wheel test show the simple and effective method has recognized 292 samples out of 300 samples correctly, so the recognition accuracy is about 97%.

(3) The proposed WTIC recognition method can help rovers to switch the terramechanics model when traversing hard or soft terrain by modulating the parameters *c* and *K* appropriately. Terrain parameters in the switched terramechanics model are identified using the least squares method, and model parameters have been successfully identified for various terrain classes, particularly for hard terrain/rock and soft sandy terrains that are similar to the main terrain types of Mars. The experiment results obtained using a Planetary Rover Prototype moving on the three terrain classes have demonstrated that the WTIC recognition, terramechanics model switching, and parameter identification are effective. The relative errors of estimated wheel–terrain interaction force with identified terrain parameters are less than 16%, 12%, and 9% for rovers traversing hard, gravel, and sandy terrain, respectively.

A vibration-based method for WTIC recognition is proposed in this paper. Previous planetary terrain analysis studies usually identify terrain types, such as ST, GT, HT, SGT, HGT, and so on [4,5,6,7,8]. When rovers move on some of these terrain types (such as ST and SGT), the WTIC is the same. It is difficult to give comprehensive and accurate statistics on the correspondence between terrain type and WTIC. Thus, these methods cannot recognize WTIC, select terramechanics models, and identify terrain parameters. In addition, terrain classification may result in misidentification between different types of terrain. It can lead to incorrect selection of terramechanics models. The WTIC recognition method proposed in this paper does not involve misidentification between different WTICs. It is more conducive to the selection of terramechanics models and the identification of terrain parameters.

In terms of terrain parameter identification, a method for identifying the terrain parameters when the rover moves on various terrains is proposed in this paper. Previous studies on terrain parameter identification (such as [16,17,18]) focused on identifying the terrain parameters of ST without identifying the terrain parameters of other types of terrain. They have not been able to switch and select terramechanics models. If the terramechanics model of wheel–ST interaction is used for terrain parameter identification of other types of terrain, it can lead to incorrect terrain parameter identification results. In terms of identifying the terrain parameters of ST, the method proposed in this paper is similar to [18]. The accuracy of identified terrain parameters is basically the same, and the relative errors of estimated wheel–terrain interaction force with identified terrain parameters are also close. Compared to previous studies on terrain parameter identification, the proposed method can achieve the terrain parameter identification of various types of terrain. It has improved the parameter identification ability.

As future work, the identification of terrain parameters in a terramechanics model in real time for planetary rovers traversing various types of terrain will be further studied. The parameters identified using this method will be used to improve the estimation accuracy of wheel–terrain interaction force in the control model for a planetary rover. And they can be used to calculate the wheel-driving torque for evaluating the terrain traversability [24]. In addition, to improve the autonomous mobility of a rover, a terrain–environment perception system of a planetary rover can be built by combining this study with vision-based terrain classification. The system can help a planetary rover construct a knowledge base of the visual terrain features and terramechanics properties and roughly estimate the terrain parameters’ value ranges in the terramechanics model for terrains that the rover will move on.

## Figures and Tables

**Figure 1 sensors-23-09752-f001:**
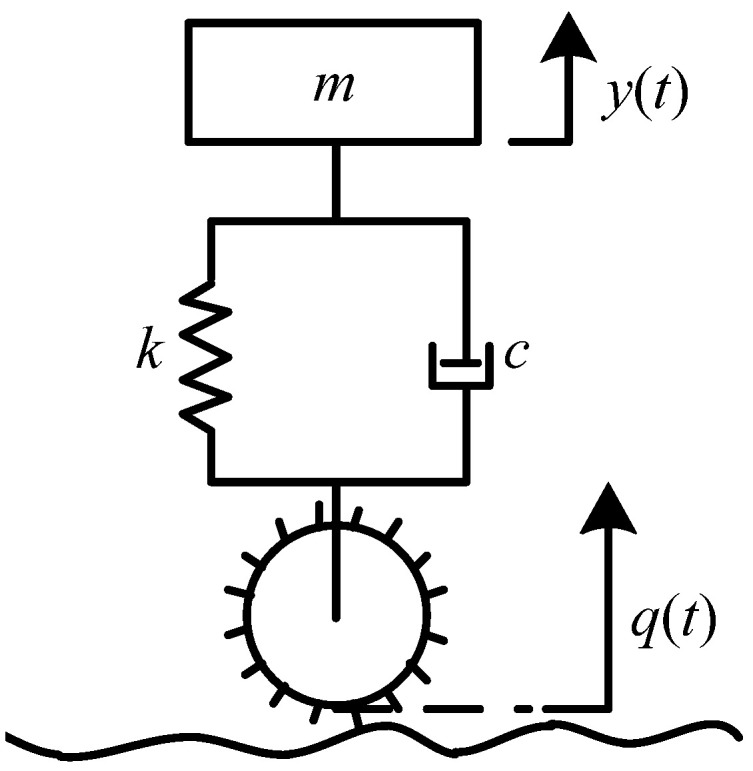
Model of a rigid lugged wheel traversing on any terrain.

**Figure 2 sensors-23-09752-f002:**
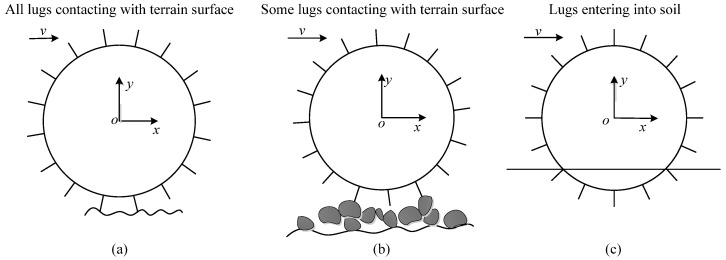
Wheel–terrain interaction classes (WTICs), with (**a**) all lugs contacting terrain surface (ALCT), (**b**) some lugs contacting terrain surface (SLCT), and (**c**) lugs entering into soil (LEIS).

**Figure 3 sensors-23-09752-f003:**
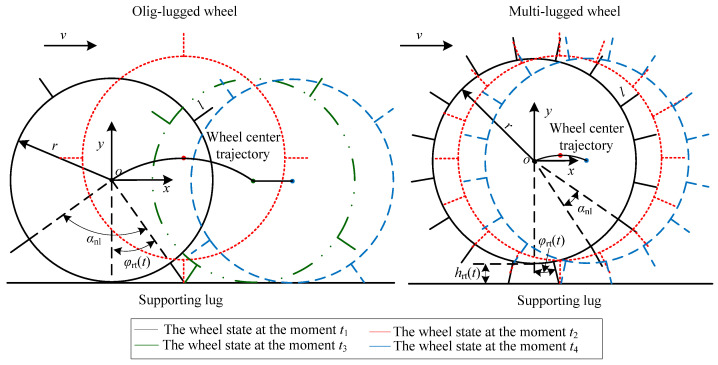
Movement phases of a wheel rotating around the supporting lug when the wheel traverses flat hard terrain.

**Figure 4 sensors-23-09752-f004:**
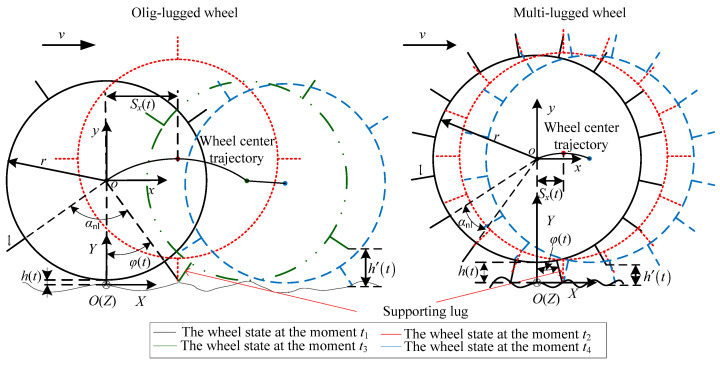
Wheel motion states of rotating around the supporting lug on natural HT.

**Figure 5 sensors-23-09752-f005:**
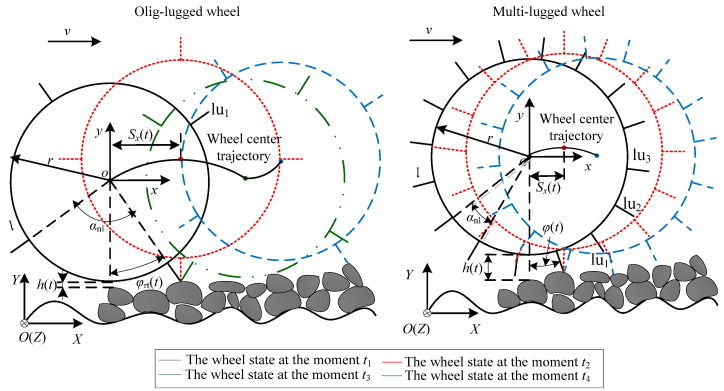
Wheel motion states of rotating around the supporting lug on GT.

**Figure 6 sensors-23-09752-f006:**
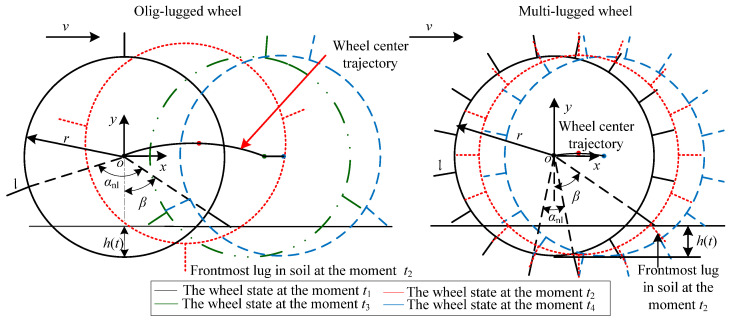
Wheel motion states when moving on sandy terrain.

**Figure 7 sensors-23-09752-f007:**
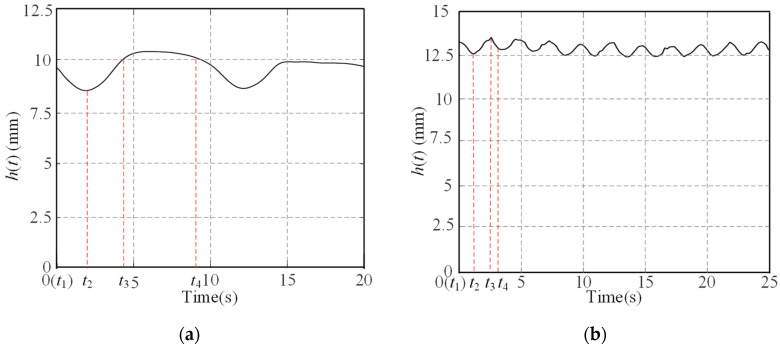
The distance h(t) between wheel rim’s lowest point and terrain surface, (**a**) when an olig-lugged wheel moves on ST, (**b**) when a multi-lugged wheel moves on ST. The black solid line represents the acceleration signal, the red dashed line represents the time point, and the black dashed line represents the coordinate grid.

**Figure 8 sensors-23-09752-f008:**
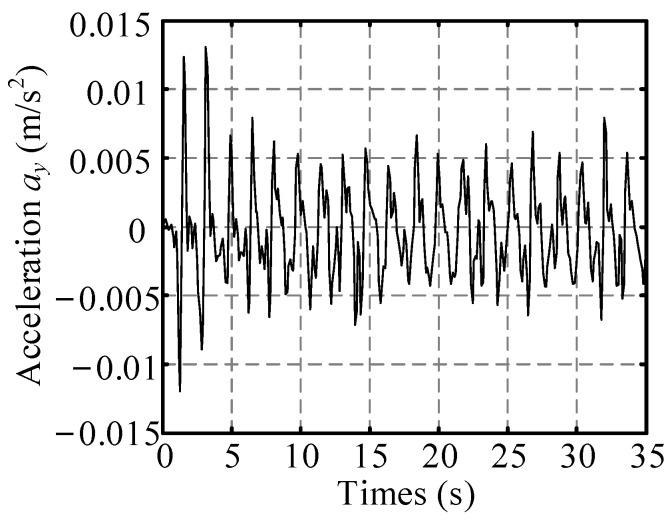
Sample of vibration acceleration when a wheel moves on ST.

**Figure 9 sensors-23-09752-f009:**
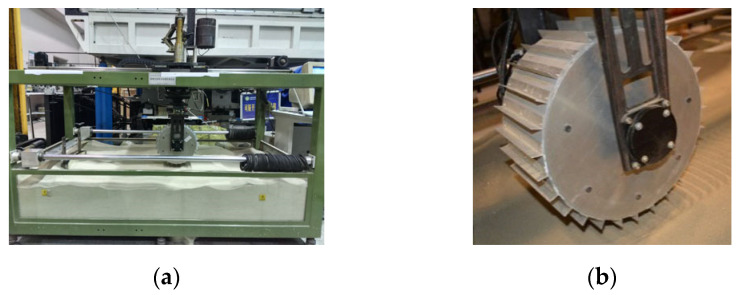
Experimental equipment, (**a**) is the wheel–soil interaction test platform and (**b**) is the experimental wheel.

**Figure 10 sensors-23-09752-f010:**
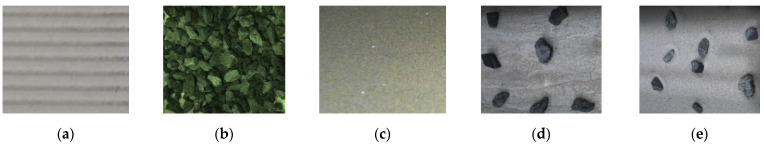
Five terrain classes: (**a**) HT, (**b**) GT, (**c**) ST, (**d**) HGT, and (**e**) SGT.

**Figure 11 sensors-23-09752-f011:**
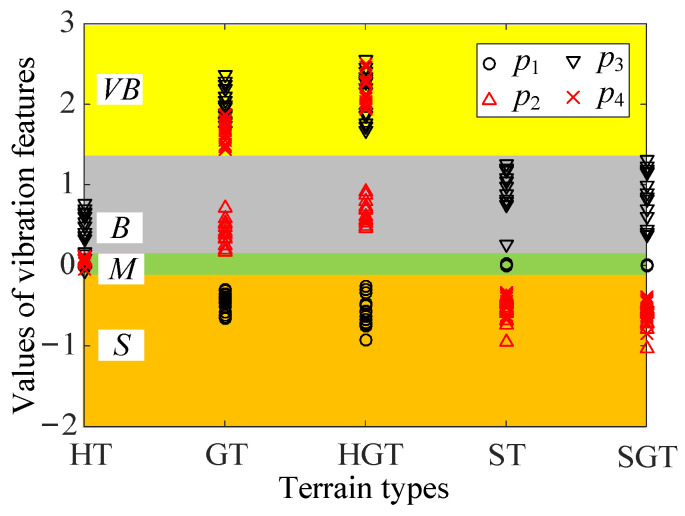
Values of vibration features for different terrain classes. Vibration features are expressed as the fuzzy set {Small (S), Medium (M), Big (B), Very-Big (VB)}.

**Figure 12 sensors-23-09752-f012:**
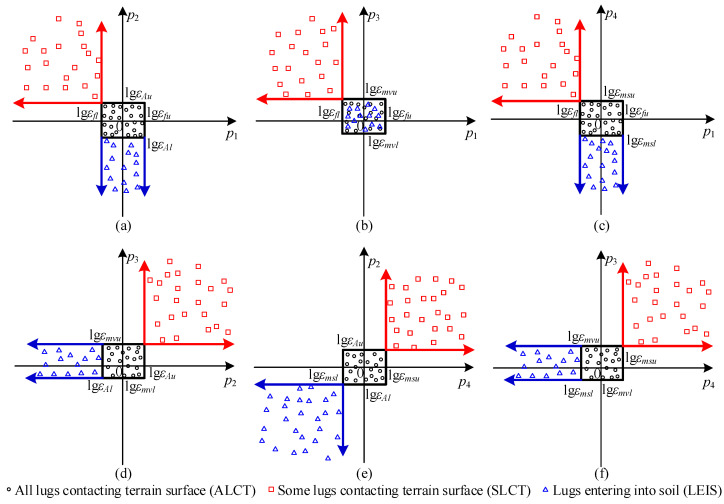
The regions of feature space formed by the vibration features for different WTICs, with (**a**) the projection in plane *p*_1_*p*_2_, (**b**) the projection in plane *p*_1_*p*_3_, (**c**) the projection in plane *p*_1_*p*_4_, (**d**) the projection in plane *p*_2_*p*_3_, (**e**) the projection in plane *p*_2_*p*_4_, (**f**) the projection in plane *p*_3_*p*_4_.

**Figure 13 sensors-23-09752-f013:**
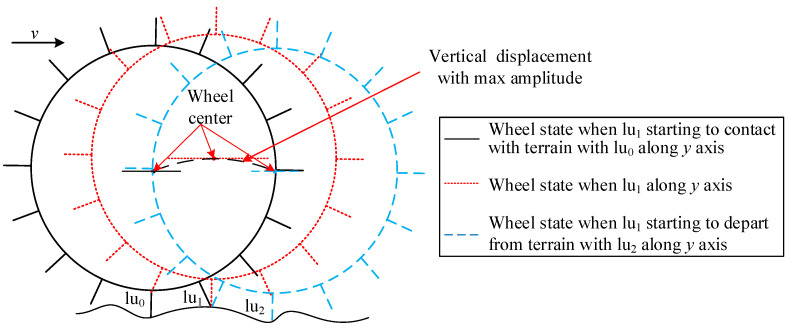
The wheel–terrain interaction case in which vibration displacement caused by lu1 has the biggest amplitude.

**Figure 14 sensors-23-09752-f014:**
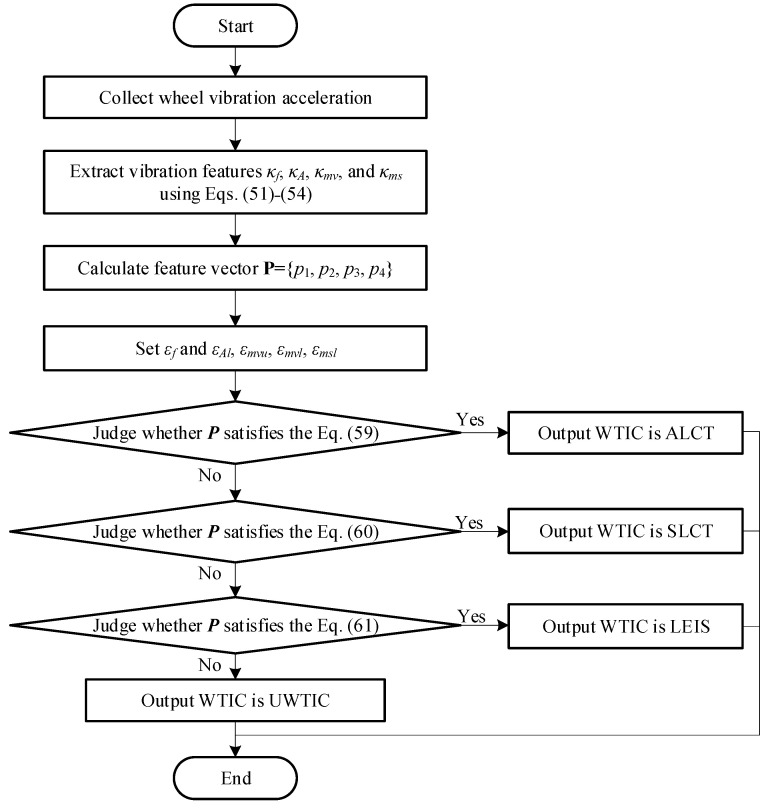
Flow chart for recognition of wheel–terrain interaction class.

**Figure 15 sensors-23-09752-f015:**
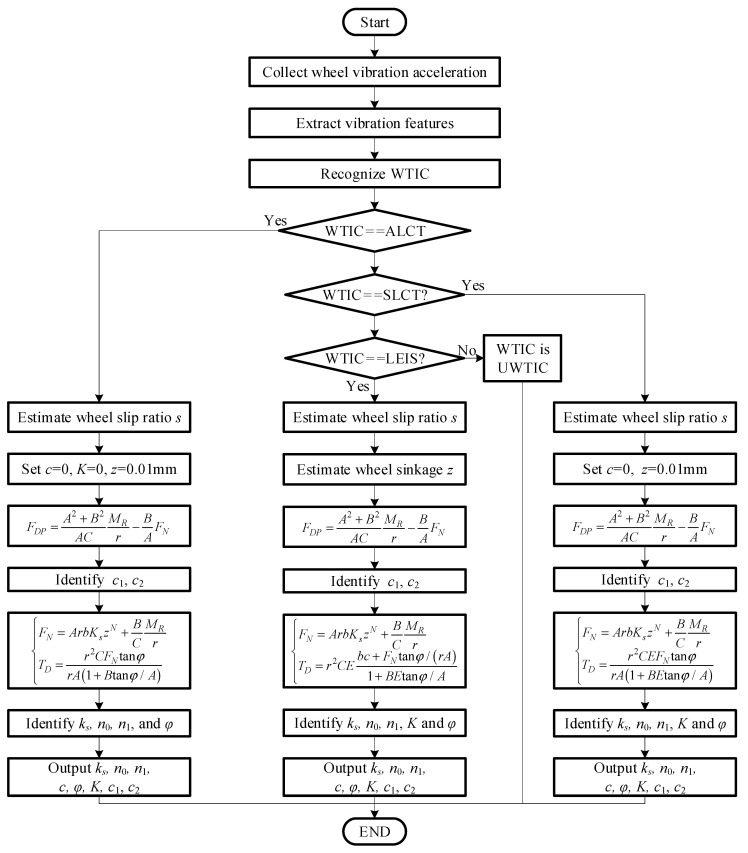
Flow chart for identifying parameters of various types of terrain.

**Figure 16 sensors-23-09752-f016:**
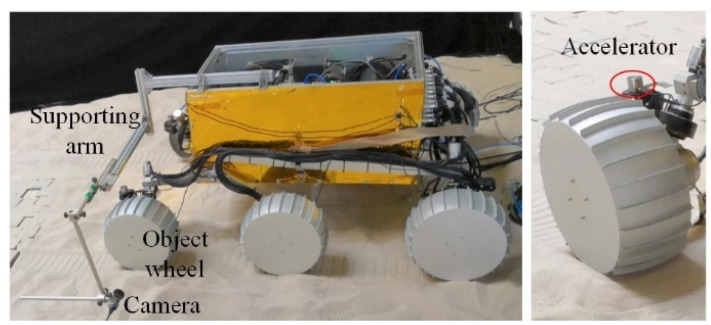
Planetary Rover Prototype.

**Figure 17 sensors-23-09752-f017:**
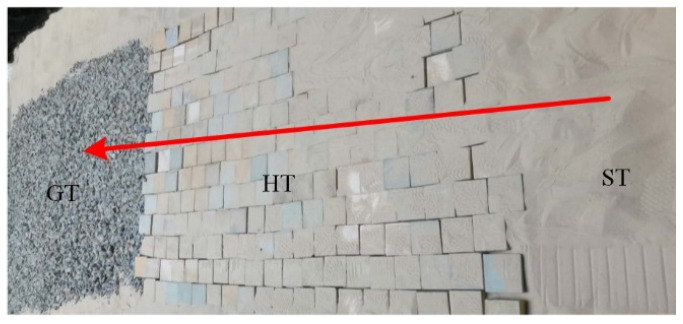
Terrain environment.

**Figure 18 sensors-23-09752-f018:**
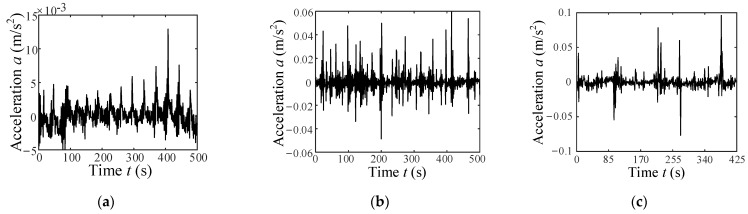
Down-sampled acceleration signals when a wheel traverses (**a**) ST, (**b**) HT, and (**c**) GT.

**Figure 19 sensors-23-09752-f019:**
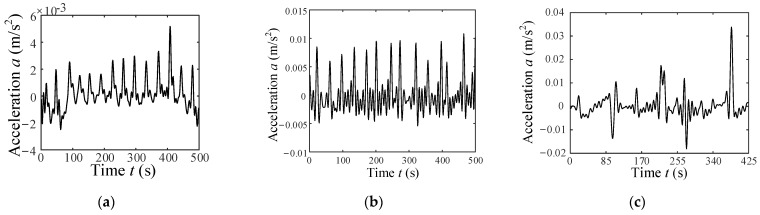
Low-pass-filtered acceleration signals when a wheel traverses (**a**) ST, (**b**) HT, and (**c**) GT.

**Figure 20 sensors-23-09752-f020:**
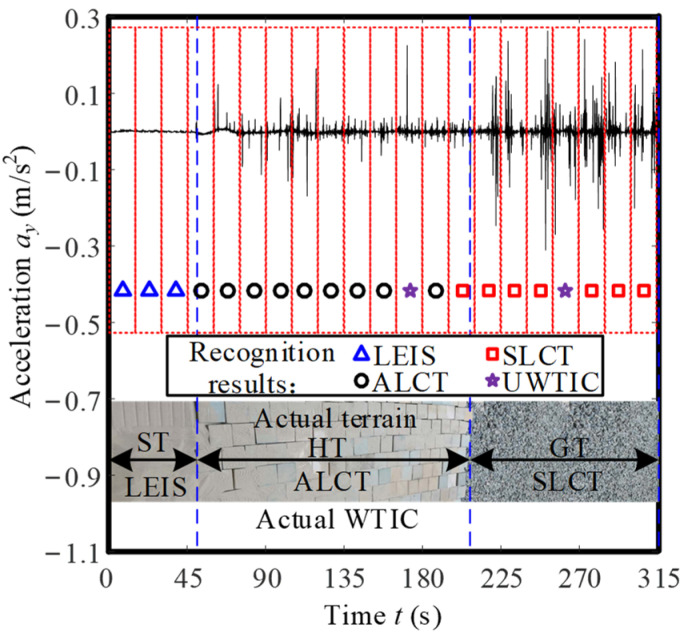
Vibration acceleration and terrain classification results for distinct single traversals of all three terrain classes.

**Figure 21 sensors-23-09752-f021:**
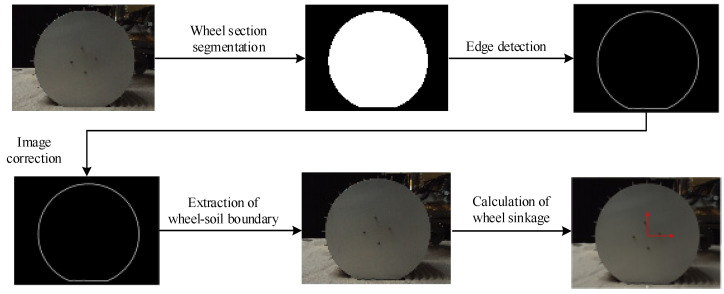
Flow chart of wheel sinkage detection.

**Figure 22 sensors-23-09752-f022:**
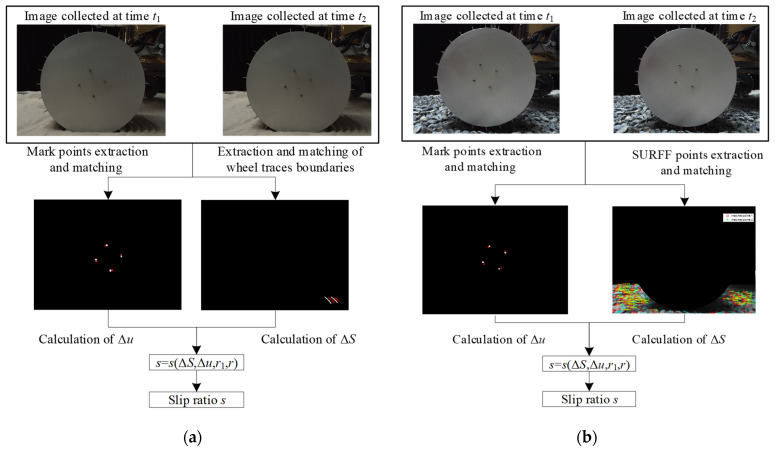
Flow charts of wheel slip ratio detection, with (**a**) for the wheel traversing ST, and (**b**) for the wheel traversing GT/HT.

**Figure 23 sensors-23-09752-f023:**
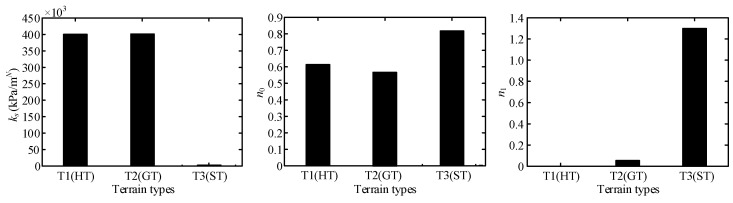
Comparison of parameters *k_s_*, *n*_0_, and *n*_1_ for three terrains.

**Figure 24 sensors-23-09752-f024:**
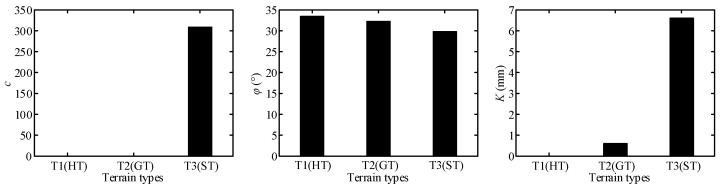
Comparison of parameters *c*, *φ*, and *K* for three terrains.

**Figure 25 sensors-23-09752-f025:**
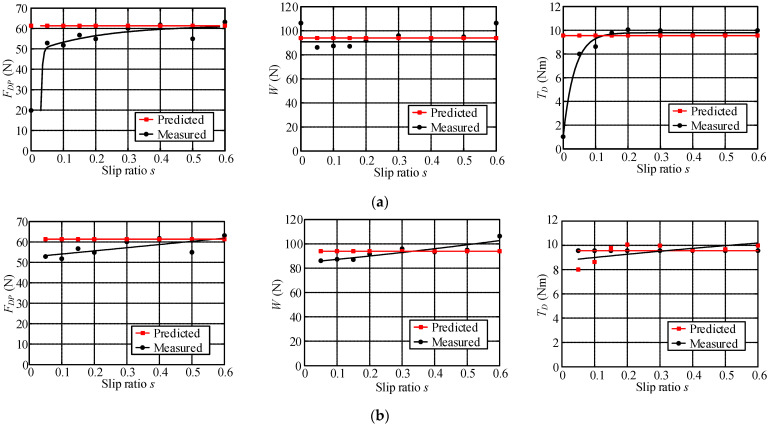
Wheel–terrain interaction force for Planetary Rover Prototype traversing HT, with (**a**) *s* = 0–0.6, and (**b**) *s* = 0.05–0.6.

**Figure 26 sensors-23-09752-f026:**
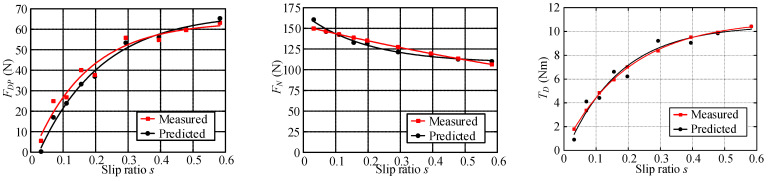
Wheel–terrain interaction force for the Planetary Rover Prototype moving on GT.

**Figure 27 sensors-23-09752-f027:**
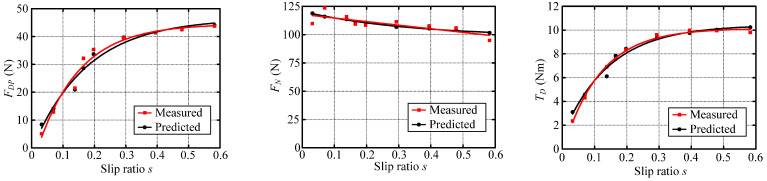
Wheel–terrain interaction force for the Planetary Rover Prototype moving on ST.

**Table 1 sensors-23-09752-t001:** Correspondence between terrain classes and the WTICs.

Terrain Class	WTIC
T1	all lugs contacting terrain surface (ALCT)
T2	some lugs contacting terrain surface (SLCT)
T3	lugs entering into soil (LEIS)

**Table 2 sensors-23-09752-t002:** The comparison of all vibration features for different WTICs.

Vibration Features	WTIC		
	ALCT	SLCT	LEIS
*p*_1_ = lg(*κ_f_*)	*M*	*S*	*M*
*p*_2_ = lg(*κ_A_*)	*M*	*B*	*S*
*p*_3_ = lg(*κ_mv_*)	*MB*	*VB*	*MB*
*p*_4_ = lg(*κ_ms_*)	*M*	*VB*	*S*
Feature vector {*p*_1_, *p*_2_, *p*_3_, *p*_4_}	{*M*, *M*, *MB*, *M*}	{*S*, *B*, *VB*, *VB*}	{*M*, *S*, *MB*, *S*}

Annotation: *MB* = *S*∪*B.*

**Table 3 sensors-23-09752-t003:** WTIC recognition results of 300 samples for the single-wheel test experiments.

Actual WTIC	Recognized WTIC			
	ALCT	SLCT	LEIS	UWTIC
ALCT (wheel–HT interaction)	58	0	0	2
SLCT (wheel–GT/HGT interaction)	0	118	0	2
LEIS (wheel–ST/SGT interaction)	0	0	116	4

**Table 4 sensors-23-09752-t004:** Correspondence between terrain class and setting values of terrain parameters for terramechanics model switching.

WTIC	Terrain Parameters	
	Soil Cohesion *c*	Shearing Deformation Modulus *K*
ALCT	0	0
SLCT	0	Without setting value
LEIS	Without setting value	Without setting value

**Table 5 sensors-23-09752-t005:** WTIC recognition results for Planetary Rover Prototype test experiments.

Actual WTIC	Recognized WTIC			
	ALCT	SLCT	LEIS	UWTIC
ALCT (wheel–HT interaction)	133	0	0	11
SLCT (wheel–GT/HGT interaction)	0	134	0	10
LEIS (wheel–ST/SGT interaction)	0	0	144	0

**Table 6 sensors-23-09752-t006:** Results of terrain parameter identification for Planetary Rover Prototype test experiments.

Terrain	*c* _1_	*c* _2_	*K_s_* (KPa/m*^N^*)	*n* _0_	*n* _1_	*c* (KPa)	*φ* (°)	*K* (mm)
T1 (HT)	0.8069	0	3.9999 × 10^5^	0.6104	0.000	0	33.34	0
T2 (GT)	0	0	4.0053 × 10^5^	0.5680	0.054	0	32.17	0.6
T3 (ST)	0.4424	−0.6379	2.4904 × 10^3^	0.8169	1.296	308.6	29.67	6.6

**Table 7 sensors-23-09752-t007:** Relative error of data fitting for *F_DP_*, *W*, and *T_D_*.

Figure Code	Figure 25a	Figure 25b	Figure 26	Figure 27
*F_DP_*	65.8%	15.2%	11.85%	8.11%
*W*	11.68%	11.68%	6.67%	7.64%
*T_D_*	84.83%	15.52%	8.46%	7.75%

## Data Availability

Data are contained within the article.

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
