# Peer review of "Vibration-Based Recognition of Wheel–Terrain Interaction for Terramechanics Model Selection and Terrain Parameter Identification for Lugged-Wheel Planetary Rovers"

_sensors, 2023, doi:10.3390/s23249752_

Round 1
Reviewer 1 Report
Comments and Suggestions for Authors
This paper proposes a speed-independent vibration-based method for WTIC recognition to switch the terra mechanics model and then identify its terrain parameters. To switch terra mechanics models, wheel-terrain interactions are divided into three classes. Three vibration models of wheels under three WTICs are built and analyzed. Vibration features in the models are extracted and nondimensionalized to be independent of wheel speed. A vibration-features-based recognition method of the WTIC is proposed. Then, the terrain parameters of the terra mechanics model corresponding to the recognized WTIC are identified. The innovation of the paper is obvious. However, many typos/errors in the equations must be corrected before the final submission.
1. Some parameters in the eqation are not defined, such as eq.(5) and eq. (18).
2. How to give the subscript in eq.(39)?
3. Interruptions appear in several figures in the paper, as shown in Figure 7, Figure 16 and 17. Please supplement or give corresponding ones.
4. Two references are missing (line 624), please add.
Reviewer 2 Report
Comments and Suggestions for Authors
The wheel-terrain interaction classes (WTICs) are usually different for rovers traversing various type terrains. The work is very interesting, and it could be accepted for publication. It would be better if the paper provides some comparsions with others' work.
Reviewer 3 Report
Comments and Suggestions for Authors
Title of the peer-reviewed manuscript: “Vibration-based Recognition of Wheel–terrain Interaction for Terramechanics Models Selection and Terrain Parameter Identification for Lugged-wheel Planetary Rovers”.
The manuscript consists of an Abstract, Keywords, an Introduction section, five main parts, a Conclusion section, list of References from 29 titles, 9 of which were published during the last 5 years. The manuscript contains 25 Figures and 9 Tables.
The studies were carried out by mathematical simulation with a experimental verification of the results on a laboratory setup.
Questions and recommendations:
1. The paper is well structured, but I would recommend that the authors also describe the structure of the paper at the end of the Introduction section after formulating the goals and objectives.
2. In the Abstract, the authors indicated that “The relative errors of estimated wheel-terrain interaction force with identified terrain parameters is less than 16%, 12%, 9% for rovers traversing hard, gravel, and sandy terrain respectively.”. However, this information is not explicitly visible in the paper. It may be worthwhile to make certain emphasis both in the main sections of the manuscript and in the Conclusion section.
3. The authors presented the texts of the programs they proposed in Tables 3 and 6. It seems to me that it would be more clear to present these algorithms in the form of block diagrams.
I have no other comments or recommendations for improving the paper.
It is difficult for me to assess the relevance and possible interest in the article of specialists in this scientific field. However, I want to point out that the paper is well organized and easy to read. I would like to congratulate the authors for a well-written article and can recommend it for publication after minor revision.
